# Whole transcriptome expression profiles in kidney samples from rats with hyperuricaemic nephropathy

**Na Li**[☯]**, Mukaram Amatjan**[☯]**, Pengke He**[☯]**, Meiwei Wu, Hengxiu Yan, Xiaoni Shao**[ORCID]*

College of Pharmacy, Southwest Minzu University, Chengdu, China

☯ These authors contributed equally to this work.
* xnshao@swun.edu.cn

**Data Availability Statement:** All raw sequence files are available from NCBI Sequence Read Archive (accession number(s) SRX16412401,

## Abstract

Hyperuricaemic nephropathy (HN) is a common clinical complication of hyperuricaemia (HUA) and poses a huge threat to human health. Hence, we aimed to prospectively investigate the dysregulated genes, pathways and networks involved in HN by performing whole transcriptome sequencing using RNA sequencing. Six kidney samples from HN group (n = 3) and a control group (n = 3) were obtained to conduct RNA sequencing. To disclose the relevant signalling pathways, we conducted the analysis of differentially expressed genes (DEGs), Gene Ontology (GO) and Kyoto Encyclopedia of Genes and Genomes (KEGG) analysis. A competitive endogenous RNA (ceRNA) network was established to reveal the interactions between lncRNAs, circRNAs, mRNAs and miRNAs and investigate the potential mechanisms of HN. Ultimately, 2250 mRNAs, 306 lncRNAs, 5 circRNAs, and 70 miRNAs were determined to be significantly differentially expressed in the HN group relative to the control group. We further authenticated 8 differentially expressed (DE)-ncRNAs by quantitative real-time polymerase chain reaction, and these findings were in accordance with the sequencing results. The analysis results evidently showed that these DE-ncRNAs were significantly enriched in pathways related to inflammatory reaction. In conclusion, HUA may generate abnormal gene expression changes and regulate signalling pathways in kidney samples. Potentially related genes and pathways involved in HN were identified.

## Introduction

Uric acid (UA) is a weak organic acid with a *pKa* of 5.75 and is the final product of exogenous purine and endogenous purine metabolism [1]. UA is principally produced by the decomposition of nucleic acids and other purine compounds metabolized by cells and purines in food through the action of enzymes and is mainly secreted by the kidney and intestines [2, 3]. HUA is universally recognized to be intimately linked to the overproduction and underexcretion of UA in invalids. The undue ingestion of high purine food and a deficiency of genetic enzymes can give rise to HUA. It is widely authenticated that exogenous sources increase serum uric acid levels, such as all meat, yeast and yeast extracts, beer and alcoholic drinks and seafood [4].

SRX16412405, SRX16412407, SRX16412409, SRX16412411,SRX16412403, SRX16412402, SRX16412406, SRX16412408, SRX16412410, SRX16412412 and SRX16412404).

**Funding:** This work was supported by the National Natural Science Foundation of China [No. 81801086], the Fundamental Research Funds for the Central Universities, Southwest Minzu University [Grant No. 2019HQZZ19], and the Natural Science Foundation of Sichuan, China [No. 2022NSFSC1574] in the form of grants to XS; by the Applied Basic Research Project of Sichuan Science Technology Department in the form of a grant to HY [No. 2021YJ0256]; and by the Innovative Research Project for Graduate students of Southwest Minzu University in 2022 in the form of grant to MA [No. ZD2022168]. The funders had no role in study design, data collection and analysis, decision to publish, or preparation of the manuscript.

**Competing interests:** The authors have declared that no competing interests exist.

There is convincing epidemiological evidence that the prevalence of HUA has substantially increased in recent decades [5]. Furthermore, HUA has been historically linked to a variety of comorbidities, including hypertension [6], metabolic syndrome [7], diabetes, chronic kidney disease [8], cardiovascular disease, obesity and gout [9, 10]. Currently, the majority of studies have demonstrated that HUA is associated with the occurrence of chronic kidney disease (CKD) and that uric acid concentration has become a standalone risk factor for kidney disease progression [11]. Even some of the articles pointed out that uric urate-lowering therapies (ULT) are potentially effective in preventing and mitigating the progression of CKD [12, 13], which is a global public health problem, and its incidence rate and mortality rate are high owing to the increased risk of developing end-stage renal disease (ESRD) and cardiovascular events [14]. But whether urate-lowering therapy has any implications for the improvement of renal dysfunction in patients with HUA remains controversial, especially for asymptomatic patients. A few reports have illustrated that urate lowering therapy has no influence on the progression of CKD in patients with asymptomatic HUA [15, 16]. Although there is no clear UA value cut-off associated with the risk for kidney damage, it appears to be more likely to emerge as UA rises [17]. Therefore, it is of great significance to perform research on the prevention and treatment of HN.

Historically, urate nephropathy has been hypothesized to result in renal damage by the deposition of intraluminal crystals in the collecting duct [18]. Numerous clinical and epidemiology studies have shown that high UA can lead to kidney damage through multiple mechanisms, including monosodium urate crystal deposition, the induction of endothelial dysfunction, renal inflammation and renal interstitial fibrosis and the activation of oxidative stress [19]. Recently, the molecular mechanisms of HN have been investigated. For example, UA can activate the protein kinase C (PKC), the mitogen-activated protein kinase (MAPK) and the cytoplasmic phospholipase A2 (cPLA2) pathways, increasing cyclooxygenase-2 (COX-2) expression, which induces vascular smooth muscle cells and tubular epithelial cells to produce monocyte chemotactic proteins (MCP-1) and platelet-derived growth factor (PDGF) [20]. MCP-1 and PDGF can directly result in kidney damage [21]. Soluble UA and UA crystals can activate the NLRP3 inflammasome with subsequent secretion of interleukin (IL)-1β to trigger innate immunity to inflammatory signals [22]. Moreover, the activation and maturation of IL-1β largely contributes to the progression of HN [23]. Although numerous specific factors have been observed, the mechanisms by which HUA leads to the development of nephropathy still need to be investigated in depth, which may facilitate a comprehensive mechanistic understanding and reveal a new therapeutic strategy for HN.

In our study, RNA sequencing (RNA-seq) was used to measure whole gene expression through RNA fragmentation, capture, sequencing, and subsequent computational analysis [24], which is a beneficial approach for the detection of common and rare transcripts and confirm other anomalous events, such as alternative splicing [25]. The total RNA transcriptome, which is the foundation of gene function and structure research, is defined as the sum of all RNAs produced by a species or specific cell under a certain functional state, and includes messenger RNAs (mRNAs) and noncoding RNAs (ncRNAs). These ncRNAs consist of microRNAs (miRNAs), long noncoding RNAs (lncRNAs) and circular RNAs (circRNAs). miRNAs, a class of single-stranded RNAs 21–22 nucleotides in length, play a vital role in the regulatory mechanisms of a variety of organisms [26]. LncRNAs are defined as a type of ncRNAs longer than 200 nucleotides in length. In light of their relative location on protein-coding transcripts, they are usually classified as intergene, intron, exon and overlapping lncRNAs [27]. CircRNA, a novel RNA that principally comprises exon sequences, is considerably different than traditional linear RNA, processes a closed-loop structure and chiefly exists in eukaryotic

transcription. Previous studies have corroborated that circRNAs probably interact with miRNA binding sites and function as miRNA sponges in different species, called ceRNAs, which can competitively bind miRNAs to regulate the expression of miRNA-targeted genes [28]. Functional interactions in ceRNA networks help to coordinate some biological processes and, when disturbed, are conducive to the pathogenesis of diseases [29]. In the last several years, increasing reports have indicated that circRNAs play a significant role in the pathological progression of many diseases [30]. Additionally, it has been shown that dysregulated expression of ncRNAs is closely associated with many common diseases, such as cerebrovascular disorders, pregnancy-related complications, diabetes and cancer [31]. This disease association stimulates our research interests and motivation for investigating the relationship between HN and ncRNAs. Through a new generation of high-throughput sequencing, almost all transcriptase sequence information of a particular tissue or organ can be obtained comprehensively and quickly. Currently, this method has been applied extensively to fundamental research, clinical diagnosis, drug research and other fields.

Although numerous pathogeneses of HN have been comprehensively identified, the cellular and molecular mechanisms underlying the ncRNAs leading to HN are not completely understood. Additionally, an increasing amount of evidence has illustrated that ncRNAs possess vital regulatory potential and participate in diversified biological processes such as cell proliferation, differentiation, invasion and apoptosis and other physiological functions [32]. Here, we established a model of HN by providing rats with a high-UA diet (HUAD), while the rats in the control group were offered normal basic feed. RNA-seq was utilized to identify DE-nRNAs. Astonishingly, we found that HUA can, to a certain degree, have an effect on gene expression in the kidney and modulate a series of signalling pathways that result in nephropathy inflammation. This study aims to provide potential biomarkers for the clinical diagnosis and treatment of nephropathy induced by HUA and to lay a solid theoretical foundation for further study of the mechanism.

## Materials and methods

### Animals

Specific pathogen-free (SPF) male Wistar rats (7–8 weeks old; weight, 180–220 g) were acquired from Vital River Laboratories. Twenty-four rats were housed in a 12 h light/dark cycle environment with a set temperature (22±2 degrees) and humidity (55±5%). Rats had unrestricted access to food and water, and the animals were acclimatized for one week before the experiment. To explore the effect of UA on the kidney, we used a previously reported method to generate a HUA model [33]. We supplied rats with a high-UA diet (HUAD) containing 2% UA and 2% potassium oxonate for periods ranging from 1 d to 12 weeks, while the rats in the control group were provided normal basic feed. After 12 weeks of modeling, the animals were executed and samples were collected. Nine rats from each group were randomly selected for serum biochemical analysis to verify the success of the model we constructed, and three kidney samples from each group were randomly chosen for whole transcriptome analysis by high-throughput sequencing. Rats were euthanatized by intraperitoneal injection with pentobarbital sodium to minimize suffering, the dosage was 40mg/kg, and the dosage was increased as needed during the experiment, then blood and kidney tissues were obtained for successive experiments. All procedures in this study were carried out in accordance with the Guide for the Care and Use of Laboratory Animals of the National Institutes of Health. The experimental protocol was approved by the Ethics Committee of College of Pharmacy, Southwest Minzu University (approval No.: 2019–08).

## Serum biochemistry analysis

Serum was obtained after coagulation (4˚C, 60 min) and centrifugation (3000 r·min$^{-1}$, 15 min, 4˚C). Clinical chemistry analysis of the serum was carried out on a Cobas C 311 biochemistry analyser (Basel, Switzerland) purchased by F. Hoffmann-La Roche, Ltd., using appropriate kits with the following parameters: UA, urea nitrogen (UREA), and serum creatinine (CREA). Serum biochemical test reagents were purchased from Mike Industrial Co., Ltd. (Chengdu, Sichuan, China). All parameter assays were performed in strict accordance with the instructions of the corresponding blood biochemical kit.

## Histopathological observation of kidney

The renal specimens were fixed in neutral-buffered 4% paraformaldehyde spending sections and embedded in paraffin wax. The tissue sections of paraffin-embedded renal tissue were stained with haematoxylin-eosin (HE) for histological analysis, and histological analysis was performed using an Olympus BX53F microscope (Olympus, Japan) equipped with a DP80 digital camera.

## High-throughput sequencing

Six kidney samples (three HNs and three controls) were designated randomly for whole transcriptome analysis by high-throughput sequencing. All analyses of RNA-Seq data were conducted with the assistance of NovelBio Bio-Pharm Technology Co., Ltd. (Shanghai, China).

## Sequencing data quality control and reference genome alignment

Initially, we used Fast-QC (http://www.bioinformatics.babraham.ac.uk/projects/fastqc/) to carry out an overall evaluation of the quality of the sequencing data, including the quality of base value distribution, quality value position distribution, GC content, PCR duplication content, and fragments per kilobase million (KMER) frequency. These evaluation metrics allowed us to gain insight into the sequencing data itself prior to mutation detection. After filtering and quality control, clean reads were obtained to analyse the genome structure. We used Hisat2 software to compare the filtered readings with a reference database (version RNOR6) [34]. Hisat2 is an efficient and fast tool for RNA-Seq data analysis, supporting genomes of any size. For miRNA mapping, the Burrows-wheeler Aligner (BWA) algorithm was utilized to compare the filtered clean reads to miRbase (http://www.mirbase.org) [35, 36] and Rfam (http://rfam.xfam.org/). The DESeq2 algorithms were applied to screen significantly DEGs with the following criteria: |log 2 FC| $\geq$ 1, false discovery rate (FDR) < 0.05. Subsequently, we used the special splicing form of circRNA in the expression process to predict the reads obtained by sequencing and found reads that simultaneously covered two exons, and the direction was opposite to that of linear RNA, to obtain the circRNA that might exist in the sequencing sample [37].

## GO and KEGG analysis

GO analysis is mainly used to query gene functions and relationships between functions and genes contained in classes on the basis of GO (http:\\www.geneontology.org), which is the functional classification of NCBI [38]. The DE-mRNA and mRNA involved in ceRNA network were annotated based on the database to obtain all the genes involved in GO. Fisher's exact test was used to calculate the p-value of each GO to screen out the significant GO with different gene enrichment under the condition of P < 0.05. In general, Fisher's exact test was utilized to classify GO and calculate FDR [39] to calibrate the P-value. The smaller the FDR was, the smaller the error was in determining the p-value. KEGG analysis was used to query the signal transduction pathways and regulatory relationships involved in genes from KEGG

(http://www.genome.jp/Kegg/) and to download pathway annotations of microarray genes. Pathway analysis is an approach to detect significant pathways of different genes according to gene annotation databases. Therefore, the critical point of pathway analysis is to have a complete database and complete pathway annotations. Initially, pathway annotation was performed on the DE-mRNAs and mRNAs involved in the ceRNA network based on the KEGG database to obtain all the pathways involved in the genes. Fisher's exact test based on hypergeometric distribution was used to calculate the P-value of each pathway, and then the significant pathways represented by different genes were screened out with $P < 0.05$ as the standard.

## CeRNA network analysis

First, the targeting relationships of significantly DE-miRNAs, DE-circRNA, DE-lncRNA, and DE-mRNA were predicted by miRanda [40] and RNAhybrid (Score < -25), respectively, and the results of the concatenation of the two prediction software were taken as the final target gene prediction results. Negative correlation association analysis was performed for miRNA-mRNA, miRNA-circRNA, miRNA-lncRAN according to differential expression type. Finally, miRNA was used as the fit point for the positive correlation joint analysis of circRNA-mRNA and lncRNA-mRNA. There are many miRNA response elements (MREs) on mRNAs that miRNAs can bind to leading to mRNA degradation or translation inhibition. Therefore, miRNAs mainly regulate mRNA expression in a negative way. On the other hand, lncRNAs and circRNAs can also adsorb miRNAs, thus affecting the regulation of miRNAs on mRNAs. These lncRNAs and circRNAs can behave as ceRNAs. The ceRNA network can reveal the patterns and functions of different ncRNAs, as well as the regulatory relationships between various ncRNAs, which bind to common miRNA binding sites to regulate gene expression through miRNA sponge mechanisms [32]. ceRNA is a novel transcriptional regulation mechanism, which suggests that lncRNAs or circulating RNAs have competitive binding forces with miRNAs through MREs, thus participating in the pathogenesis of HN [41]. Combined with the prediction of target genes, the lncRNA/circRNA, we ensured that the miRNA and mRNA of the joint analysis results for the same lncRNA/circRNA were similarly up and downregulated. The construction of ceRNA joint networks took advantage of Cytoscape [42].

## Quantitative real-time polymerase chain reaction validation

In this validation, we randomly selected 10 kidney samples from the HN group (n = 5) and control group (n = 5) to verify the 8 significantly DE-ncRNAs by quantitative real-time polymerase chain reaction validation (qRT-PCR). Total RNA from the kidney samples of rats was extracted using an Animal Total RNA Isolation Kit (Foregene, Chengdu, China), and a K2800 nucleic acid analyser (Beijing Kaiao Technology Development Co., Ltd) was used to detect the concentration and purity of RNA. MiRNA L-RT Enzyme mix and Servicebio RT Enzyme Mix were used to synthesize the complementary deoxyribonucleic acid (cDNA) of miRNAs and lncRNAs, respectively. And 5×All-In-One MasterMix (with AccuRT Genomic DNA Removal kit) was used to for reverse transcription of mRNAs. qRT-PCR was performed in a reaction volume of 20 μl, including 10.0 μl 2× qPCR MasterMix (Abm, Vancouver, Canada), 1.2 μl 7.5 μM gene primer Mix (Sangon Biotech, Shanghai, China), 2.0 μl cDNA, and 6.8 μl ddH2O. qRT-PCR was completed with the Automatic Medical PCR Analysis System purchased from Shanghai Hongshi Medical Technology Co., Ltd. PCR amplification was conducted at 95˚C for 10 min, followed by 40 cycles of 95˚C (10 s) and 60˚C (30 s), ultimately, the temperature rose from 60˚C to 95˚C by 0.3˚C every 15 s. Three pores were prepared for each reverse transcriptome. The relative fold changes of ncRNA expression were calculated using the formula 2 −ΔΔCt [43]. And the qRT-PCR primer sequences are listed in S1 Table.

## Statistical analysis

Data in this study are displayed as mean±standard error of mean (SEM). Student' s t-test was applied to verify the differences between two groups. The association between samples was analysis by using Pearson correlation analysis. GraphPad 5.0 was adopted to draw graphs. P-value < 0.05 was regarded as statistical significance.

## Data availability

The raw sequence data in this study have been deposited into the NCBI Sequence Read Archive (http://trace.ncbi.nlm.nih.gov/Traces/sra/sra.cgi?view=studies), and the accession numbers of the six SRA samples for RNA-seq are as follows: SRX16412401, SRX16412405, SRX16412407, SRX16412409, SRX16412411 and SRX16412403. And numbers of the six SRA samples for miRNA-seq are as follows: SRX16412402, SRX16412406, SRX16412408, SRX16412410, SRX16412412 and SRX16412404.

# Results

## Animal characteristics

The characteristics of rats fed HUAD are shown. The serum levels of UA, urea nitrogen (UREA), and serum creatinine (CREA) were measured using clinical biochemistry analysis. As shown in Fig 1, there was a threefold increase in the serum UA concentration, from 102.5 ±8.52 μmol/L in control rats to 246.6±29.42 μmol/L in HN rats, which indicated that we successfully created a HUA rat model (Fig 1A). The UREA level also increased from 6.93±0.45 mMol/L in control rats to 19.93±1.286 mMol/L in HN rats (Fig 1B). Under pathophysiological conditions, drastic UREA level increases provide key information on renal function and the diagnosis of various kidney disorders [44]. Likewise, the CREA level in the control group was 34.22±1.54 μmol/L, whereas in HN rats, it increased to 79.44±4.37 μmol/L, which suggests the presence of kidney dysfunction induced by HUAD (Fig 1C).

Histopathological examination showed that the kidney of rats in HN group had developed pathological atrophy, with radiating patterns in the medulla and blurred cortical margins (Fig 1D and 1E). HE staining showed that, compared with the control group, the kidney tissues of the HN rats revealed obvious inflammatory cell infiltration, tubular epithelial cell necrosis, severe tubular dilatation, glomerular hyperplasia and uric acid crystals in the kidney tissues (Fig 1F and 1G). These pathological characteristics are analogous to those of HN in humans [5]. Thus, these histological findings showed that HUAD resulted in nephropathy.

## Whole-transcriptome sequencing data

A total of 510.27 M raw reads (76.54 G bases) were obtained by whole transcriptome sequencing in a lamp-specific library with ribosomal RNA removed. After filtration and quality management, we obtained 497.58 M reads (74.55 bases), with an average of 82.94 M reads and 12.42 G bases per sample; the overall sample had an average GC content of 48.34% (Table 1). The abovementioned information suggests that we obtained high-quality RNA-seq data. Every sample was subjected to separate comparisons with the reference genome. Consequently, 84.8% of clean reads mapped to the rat reference genome, and the alignment rates at intergenic, intron, and exonic regions were 10.15%, 27.14% and 62.78%, respectively. From the Pearson correlation (Fig 2A) and PCA chart (Fig 2B), which demonstrates that there is a correlation between the genetic expression of each sample.

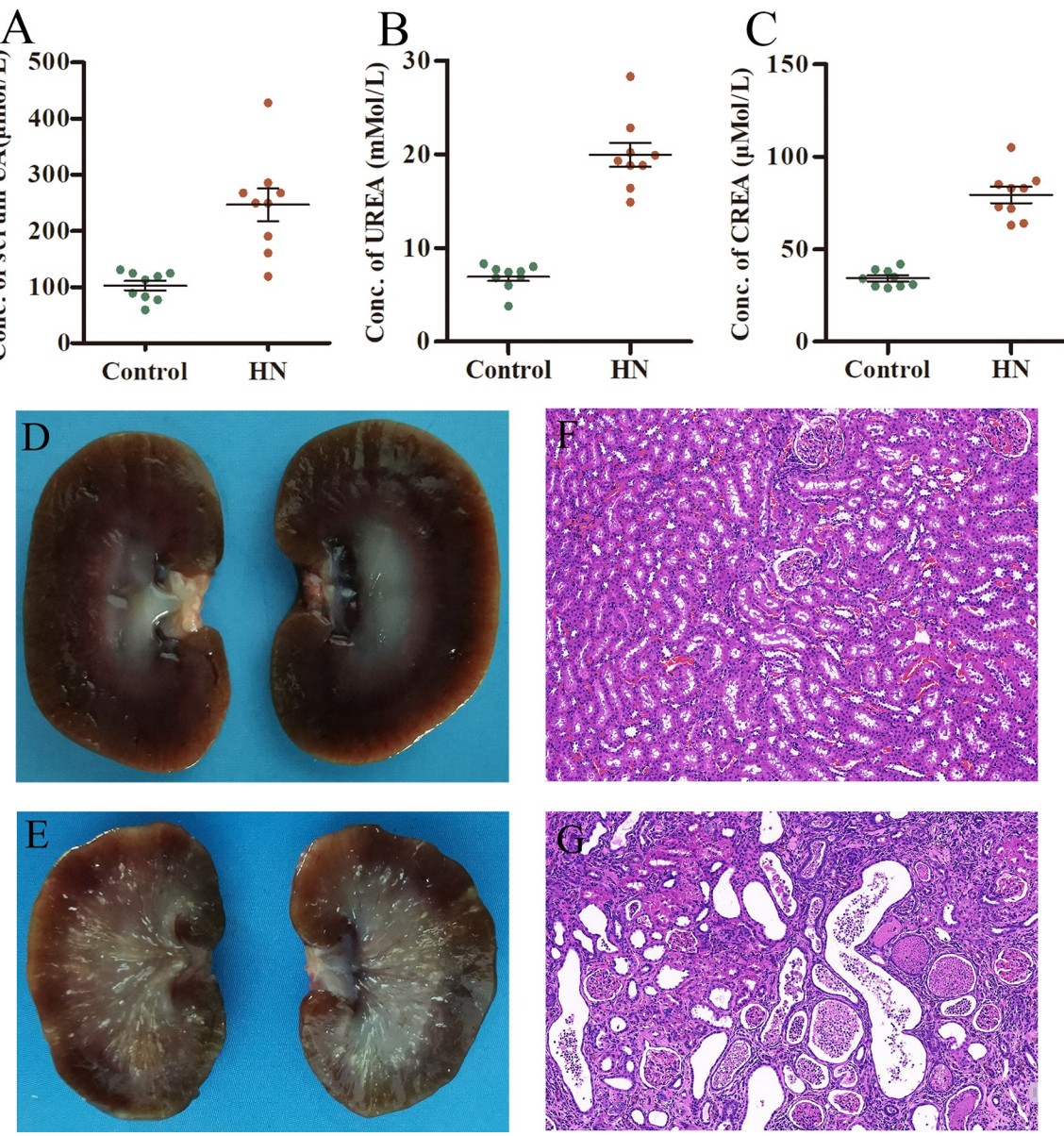

**Fig 1. Characteristics of rats fed a normal or HUAD.** (A) There were differences in the serum UA concentration, (B) serum UREA concentration and (C) serum UREA between HN rats (n = 9) and normal control rats (n = 9). For the comparison of quantitative data between groups, Student's t-test was used. Differences in means were considered statistically significant at *P < 0.05. (D) Kidney longitudinal section photo of control rats. (E) Kidney longitudinal section photo of HN rats. (F) Kidney microscopic image of control rats. (G) Kidney microscopic image of HN rats. HN, hyperuricaemic nephropathy.

**Table 1. Summary of RNA-seq data.**

| Sample | Raw reads | Clean reads | Ratio (%) | GC (%) | Mapped ratio (%) | Exonic (%) | Intronic (%) | Intergenic (%) |
|---|---|---|---|---|---|---|---|---|
| HN1 | 83637622 | 80438950 | 96.18 | 48.03 | 85.6 | 58.63 | 31.36 | 10.09 |
| HN2 | 91820872 | 88252442 | 96.11 | 47.53 | 85.8 | 57.80 | 32.21 | 10.05 |
| HN3 | 90145696 | 86926294 | 96.43 | 48.20 | 86.8 | 60.02 | 30.04 | 10.02 |
| Control1 | 86975464 | 86290154 | 99.21 | 49.22 | 83.5 | 71.08 | 19.25 | 9.75 |
| Control2 | 85164030 | 84409808 | 99.11 | 48.22 | 83.7 | 64.41 | 25.02 | 10.66 |
| Control3 | 72529806 | 71263642 | 98.25 | 48.85 | 83.6 | 64.74 | 24.99 | 10.34 |
| Total | 510.27M | 497.58 M | / | / | / | / | / | / |
| Average | 85.04 M | 82.94 M | 97.54 | 48.34 | 84.8 | 62.78 | 27.14 | 10.15 |

**Table 2. Summary of circRNA prediction.**

| Sample | Count | Max length | Min length | Average length |
|---|---|---|---|---|
| HN1 | 4326 | 2496257 | 103 | 36043 |
| HN2 | 4861 | 2496257 | 101 | 35471 |
| HN3 | 5297 | 2478665 | 102 | 34497 |
| Control1 | 5359 | 2478665 | 101 | 34181 |
| Control2 | 4631 | 2478665 | 102 | 35030 |
| Control3 | 6055 | 2479877 | 102 | 30581 |
| Average | 5088 | 2484731 | 102 | 34301 |

## Identification of DE-ncRNAs

The gene FPKM in each sample were calculated to conduct differential gene analysis [|log 2 (fold change) | value > 1, FDR < 0.05 was regarded as the threshold]. In total, we identified 2631 DE-transcripts, which included 1893 upregulated ncRNAs [log2 (fold change) >1, FDR < 0.05] and 738 downregulated ncRNAs [log2 (fold change) < -1, FDR < 0.05] (Fig 3A). Hierarchical clustering analysis of DE-transcripts indicated that these transcripts had obviously disparate expression between the HN kidney samples and control samples (Fig 3B). Compared with the control group, 2250 mRNAs, 70 miRNAs, 306 lncRNAs and 5 circRNAs were DE in the HN group. A total of 1684 mRNAs, 50 miRNAs, 156 lncRNAs and 3 circRNAs were upregulated, while 566 mRNAs, 20 miRNAs, 150 lncRNAs and 2 circRNAs were downregulated in the HN group (Fig 3C).

## Screening and analysis of DE-mRNAs

Based on established thresholds [log2 (fold change) > 1, FDR < 0.05], a total of 2250 DE-mRNAs were screened, including 1684 upregulated genes [log2 (fold change) >1, FDR < 0.05] and 566 downregulated genes [log2 (fold change) < -1 FDR < 0.05] (Fig 4A).

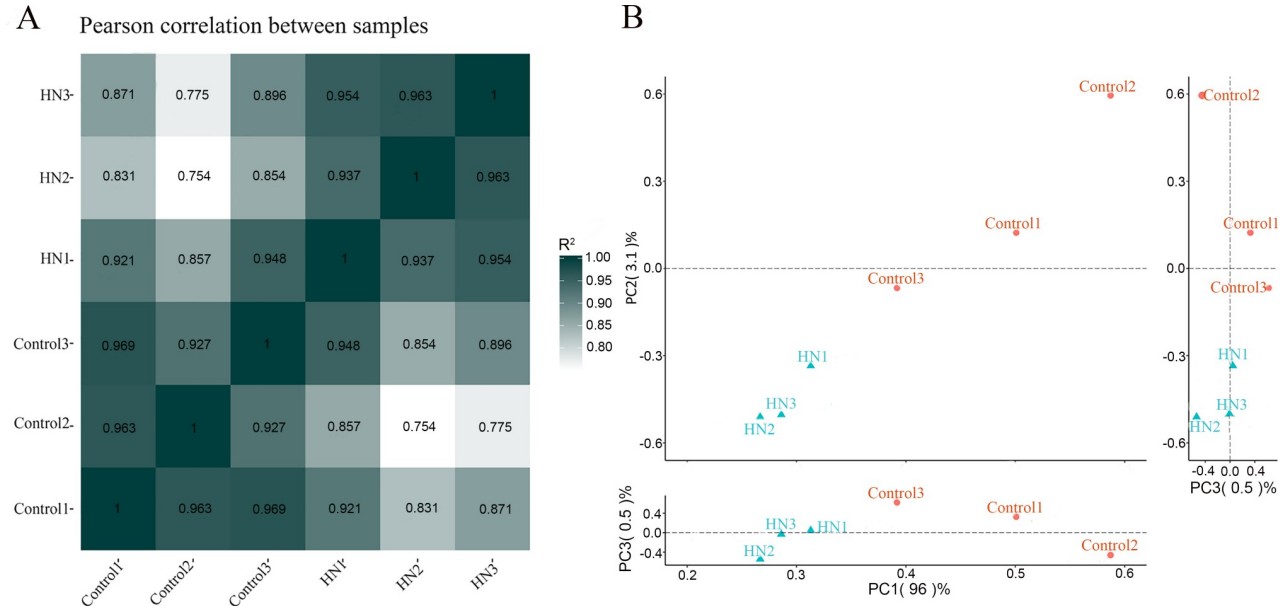

**Fig 2. The Pearson correlation coefficient between samples and principal component analysis.** (A) Pearson correlation coefficient (PCC) between samples; the closer the square value of R is to 1, the higher the correlation between samples. (B) Principal Components Analysis (PCA).

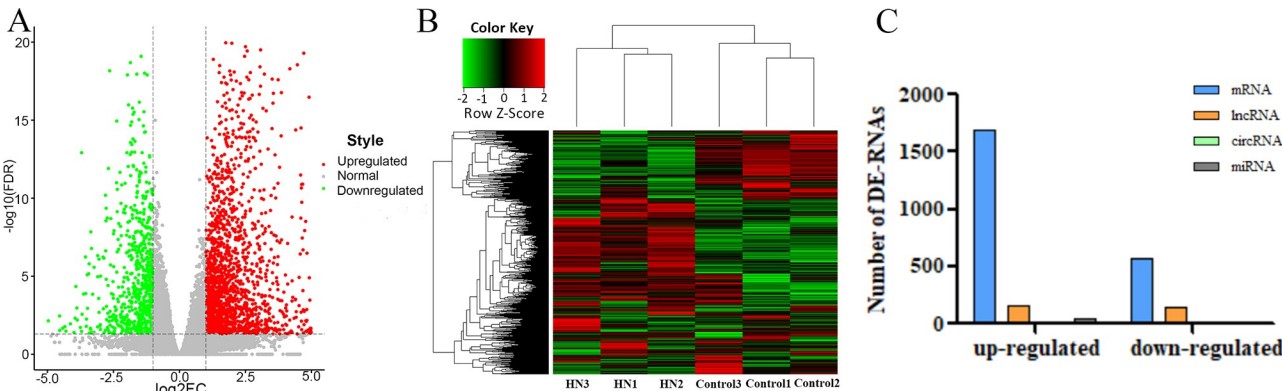

**Fig 3. Characteristics of all DE-transcripts.** (A) Volcano plot of all DE-transcripts between the rats suffering from HN and normal control rats. Red nodes refer to transcripts that were upregulated [log2 (fold change) > 1, FDR < 0.05], while green nodes represent transcripts that were downregulated [log2 (fold change) < -1, FDR < 0.05]. Grey points indicate normally expressed transcripts. (B) The DE-transcripts in HN were analysed via hierarchical clustering. The closer to a red colour, the higher of expression. In contrast, a green denotes downregulated expression profiles. (C) Characteristics of DE-transcripts. Blue, orange, green and grey colours denote mRNAs, lncRNAs, circRNAs and miRNAs, respectively. HN, hyperuricaemic nephropathy; FDR, false discovery rate.

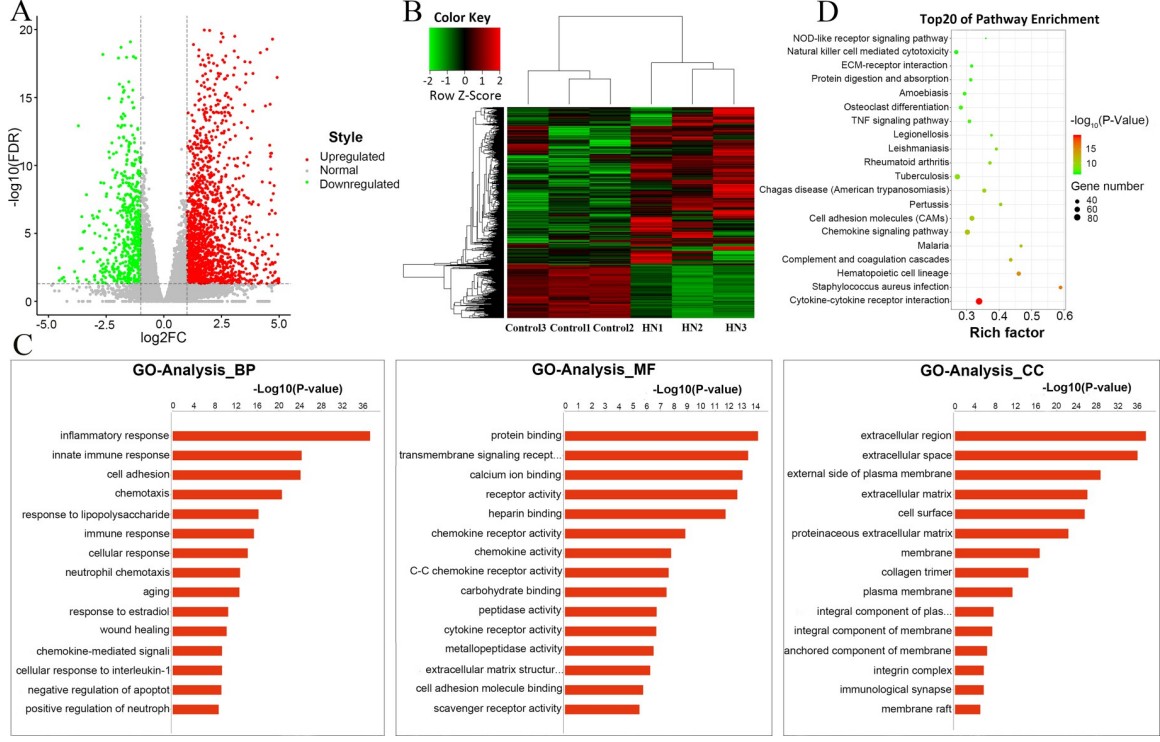

**Fig 4. The identification and further analysis of DEGs.** (A) Volcano plot for the comparison between HN rats and the rats of the control group. Red points indicate upregulated mRNAs [log2 (fold change) > 1, FDR < 0.05], while green nodes indicate downregulated mRNAs [log2 (fold change) > -1, FDR < 0.05]. Grey dots refer to normally expressed mRNAs. (B) Heatmap analysis was used to determine the differential expression between the HN group and the control group. Each row indicates a single mRNA, and each column indicates a tissue sample. (C) Gene Ontology analysis of DEGs, including MF, CC and BP classification. The horizontal axis is the enrichment value of functional degree, and the vertical axis is the entry name corresponding to GO in the Gene Ontology database. (D) Kyoto Encyclopedia of Genes and Genomes pathway analysis of DEGs. The size of the bubbles indicates the number of genes involved in pathways; bubble colour indicates the P-value. DEGs, differentially expressed genes; HN, hyperuricaemic nephropathy; mRNA, message RNA; FDR, false discovery rate; GO, Gene Ontology; MF, molecular function; CC, cellular component; BP, biological process.

According to the heatmaps of DE-mRNA clusters, we can easily conclude that these DEGs had remarkably different expression patterns in the HN case than in the control samples (Fig 4B), which demonstrates that the DE-mRNAs we screened had obvious differences in characteristics. The top 5 upregulated genes included Mmp7, Cxcl6, Tdo2, Chst5, and Nos2, while the top 5 downregulated genes included Klk1c6, Slc22a13, Nhlh2, Cyp3a71-ps, and Scgb1c. The specific information detailing DE-mRNA is described in S2 Table. GO and KEGG analyses were carried out to further study the function of DEGs. The top 30 GO terms are listed in Fig 4C. Biological processes (BPs) that were substantially related to these DEGs included inflammatory response, innate immune response, cell adhesion, chemotaxis and lipopolysaccharide response. The cellular components (CCs) that were remarkably associated with these genes consisted of the extracellular region, the extracellular space, the external side of plasma membrane, the extracellular matrix and cell surface. Molecular functions (MFs) that were significantly associated included protein binding, transmembrane signalling receptor activity, calcium ion binding, receptor activity and heparin binding. Based on the hierarchical structure of GO, the mutual regulation and subordination between all GO terms were organized into a database. Through the construction of a functional relationship network, we can easily summarize the functional groups affected by the experiment, as well as the internal subordination of significant functions. Here, we chose the significant GO terms ($P < 0.01$) in the BP level GO category of the GO analysis to construct the functional regulation network (Fig 5A). And S3 Table listed all meaningful GO annotations. Subsequently, analysis of KEGG pathway enrichment suggested that the DEGs between the HN group and the control group were significantly associated with 90 KEGG pathways, including PATH:04060 (cytokine-cytokine receptor interaction), PATH:05150 (Staphylococcus aureus infection), PATH:04640 (haematopoietic cell lineage, complement), and PATH:04610 (coagulation cascades). The most enriched pathway was the cytokine-cytokine receptor interaction. The top 20 KEGG pathways are illustrated in Fig 4D, and all KEGG pathways were shown in S4 Table. Similarly, the relationships between all pathways were arranged into a database, and we constructed a signalling pathway regulation network based on selecting the pathway terms ($P < 0.05$) of analysis. We sought to uncover the signaling relationships between pathways and to preliminarily explore the potential core pathways affected by the experiment and the regulatory mechanisms between signaling pathways (Fig 5B).

## Screening and analysis of DE-ncRNAs

Similarly, in our study, 306 DE-lncRNAs were identified, including 156 upregulated lncRNAs [log2 (fold change) > 1, FDR < 0.05] and 150 downregulated lncRNAs [log2 (fold change) < -1, FDR < 0.05] (Fig 6A). Heatmaps of DE-lncRNA clusters are shown in Fig 6B. We individually predicted 4326, 4861, 5297, 5359, 4631, and 6055 from 6 sequencing samples, with an average length of 3430 (Table 2). A total of 5 DE-circRNAs were obtained, of which 3 were upregulated and 2 were downregulated. S5 and S6 Tables provided detailed information on DE-lncRNAs and DE-circRNAs, respectively.

Regarding miRNAs, we acquired 74.94 M reads in total by constructing a miRNA library and performing sequencing. After filtering and quality control, 53.09 M clean reads were obtained, and the average alignment rate with the reference genome was 61.42%. Similarly, we identified 70 DE-miRNAs [|log 2 (fold change) | value > 1, FDR < 0.05], including 50 upregulated miRNAs [log2 (fold change) >1, FDR < 0.05] and 20 downregulated miRNAs [log2 (fold change) < -1, FDR < 0.05]. The DE-miRNAs in the volcano plot (Fig 6C) and cluster analysis (Fig 6D) are performed. And further information about DE-miRNA was mentioned in the S7 Table.

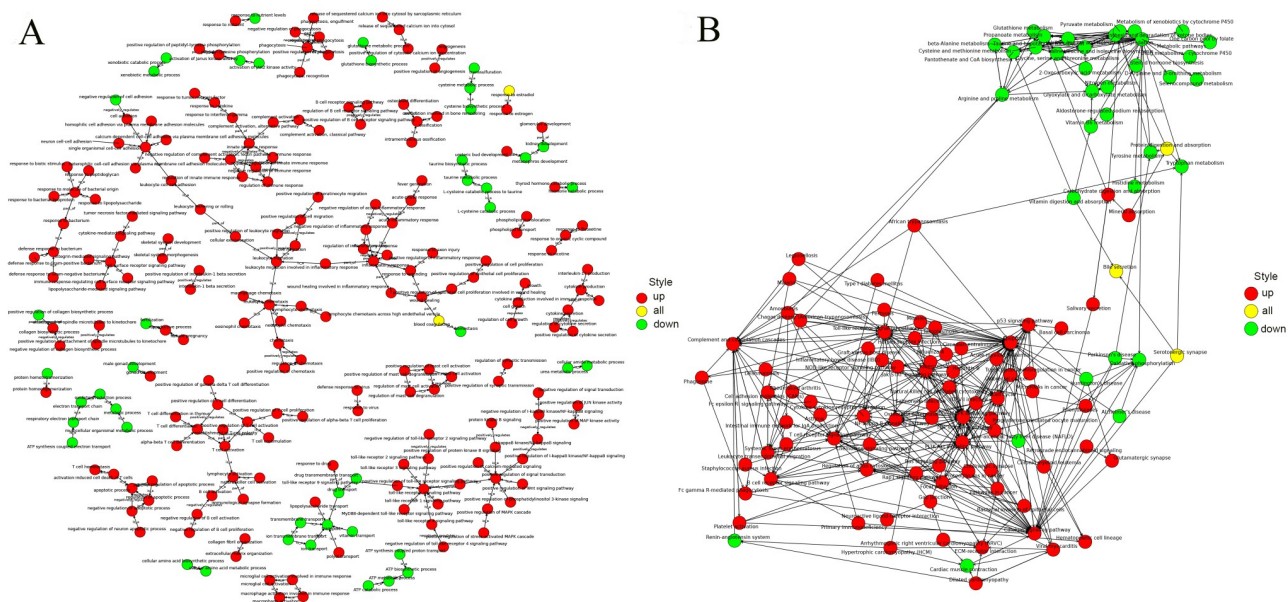

**Fig 5. The GO-Trees and the Pathway-Act-Network.** (A) Hierarchical tree diagram among salient functions and GO trees of significant GO terms (P < 0.01). Green indicates significant GO terms in which downregulated genes are involved, red indicates significant GO terms in which upregulated genes are involved, and yellow indicates significant GO terms in which both upregulated and downregulated genes are involved. (B) Signalling pathway regulatory network. The upregulated and downregulated pathway terms (P < 0.05). Green indicates the significant pathway term involved in downregulated genes, red indicates the significant pathway term involved in upregulated genes, and yellow indicates the significant pathway term involved in both upregulated and downregulated genes.

## Target prediction

1308 miRNA-mRNA interaction relationships (including 780 DE-mRNAs and 63 DE-miR-NAs), 515 miRNA-lncRNA interaction relationships (including 217 DE-lncRNAs and 58 DE-miRNAs) and 122 miRNA-circRNA interaction relationships (including 5 DE-circRNAs and 67DE-miRNAs) were identified. The specific method of circRNA prediction is as follows, first based on sequenced Clean Reads, applying the ACFS2 algorithm for circRNA prediction of samples. Based on the BWA algorithm for comparison of sequencing results, Clean Reads that do not match to the reference genome can be used for prediction of cyclic RNAs. Junction Reads of "head-to-tail" type are first identified and then scored by the MaxEntScan33 algorithm, and those with a score of ≥10 were retained. Reads with a score of ≥10 were rematched to the trans-shear site region of the alternative circular RNA, and those that can be matched (at least 6 bases) can be used to determine and calculate the expression of the circRNA. MiRNAs can cause gene silencing by binding to target genes, and in cells, miRNAs alter gene expression mainly through negative regulation [45]. Prediction of lncRNA-mRNA interaction pairs showed that there were 491 miRNA-mRNA negative pairs [including 390 DE-mRNAs and 50 DE-miRNAs], 243 miRNA-lncRNA negative pairs [including 152 DE-lncRNAs and 51 DE-miRNAs] and 43 miRNA-circRNA negative pairs [including 5 DE-circRNAs and 39 DE-miRNAs].

## The lncRNA-miRNA-mRNA ceRNA network map and analysis

Previous studies have found that mRNAs can be influenced by lncRNAs through miRNAs, through which a lncRNA-associated ceRNA network can be established [46]. The intersecting DE-lncRNAs, miRNAs, and mRNAs described above were adopted to build a ceRNA network

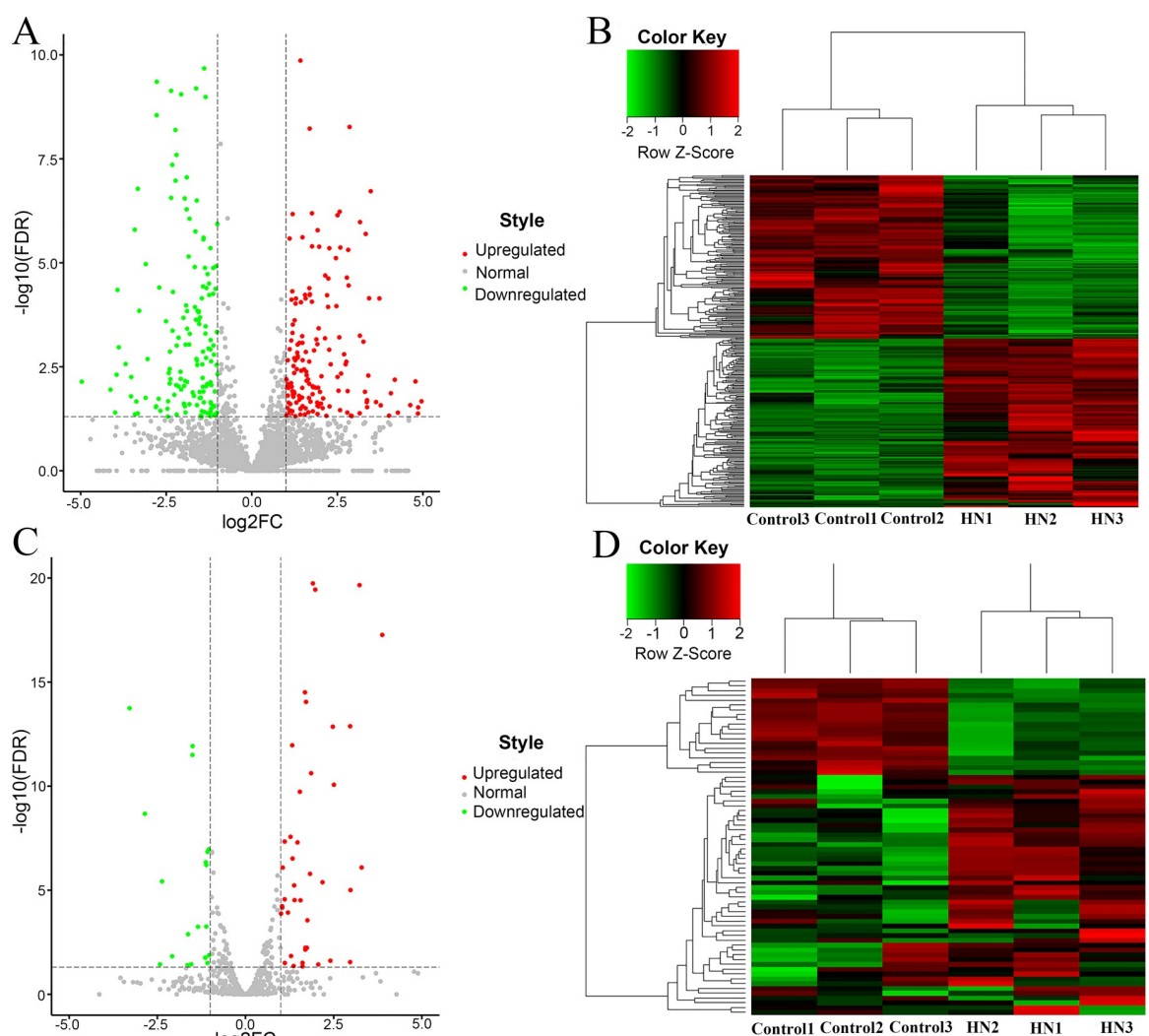

**Fig 6. DE-lncRNAs and miRNAs in the renal HN rat model.** (A) Volcano plot of DE-lncRNAs and miRNAs (C) between rats suffering from HN and normal control rats. Red nodes refer to lncRNAs that were upregulated [log2 (fold change) > 1, FDR < 0.05], while green nodes represent lncRNAs that were downregulated [log2 (fold change) < -1, FDR < 0.05]. Grey points indicate the lncRNAs and miRNAs expressed normally. (B) Hierarchical clustering of lncRNAs and miRNAs (D) shows the differential expression between the HN group and the control group. The closer to red, the higher the expression level. In contrast, green denotes downregulated expression profiles. LncRNA, long noncoding RNA; miRNA, microRNA; HN, hyperuricaemic nephropathy; FDR, false discovery rate.

in HN. This lncRNA-miRNA-mRNA ceRNA network map was made up of 576 ceRNAs, including 42 DE-miRNAs, 386 DE-mRNA, and 148 DE-lncRNAs (S8 Table). We selected the 5 DE-miRNAs with the most interactions (including 3 upregulated miRNAs, rno-miR-351-5p, rno-miR-214-3p, rno-miR-212-5p and 2 downregulated miRNAs, rno-miR-709, rno-miR-760-5p) to determine which had the greatest likelihood of being involved in HN. According to the results, we discovered that a majority of miRNAs were regulated by numerous mRNAs and lncRNAs (Fig 7A). Examples include Lpcat1/Scn2b (up)- rno-miR-709 (down)-LOC103690809/LOC102547703/LOC102548523 (up), Clcf1/Ptafr (up)- rno-miR-760-5p (down)- LOC100909928/LOC103692000/LOC103691582 (up), Usp2/Cpxm2 (down)-rno-miR-351-5p (up)- LOC103692475/LOC10369444 (down), and Tmem72 (down)- rno-miR-351-5p/rno-miR-212-5p (up)- LOC100909941 (down). To further unravel the functional

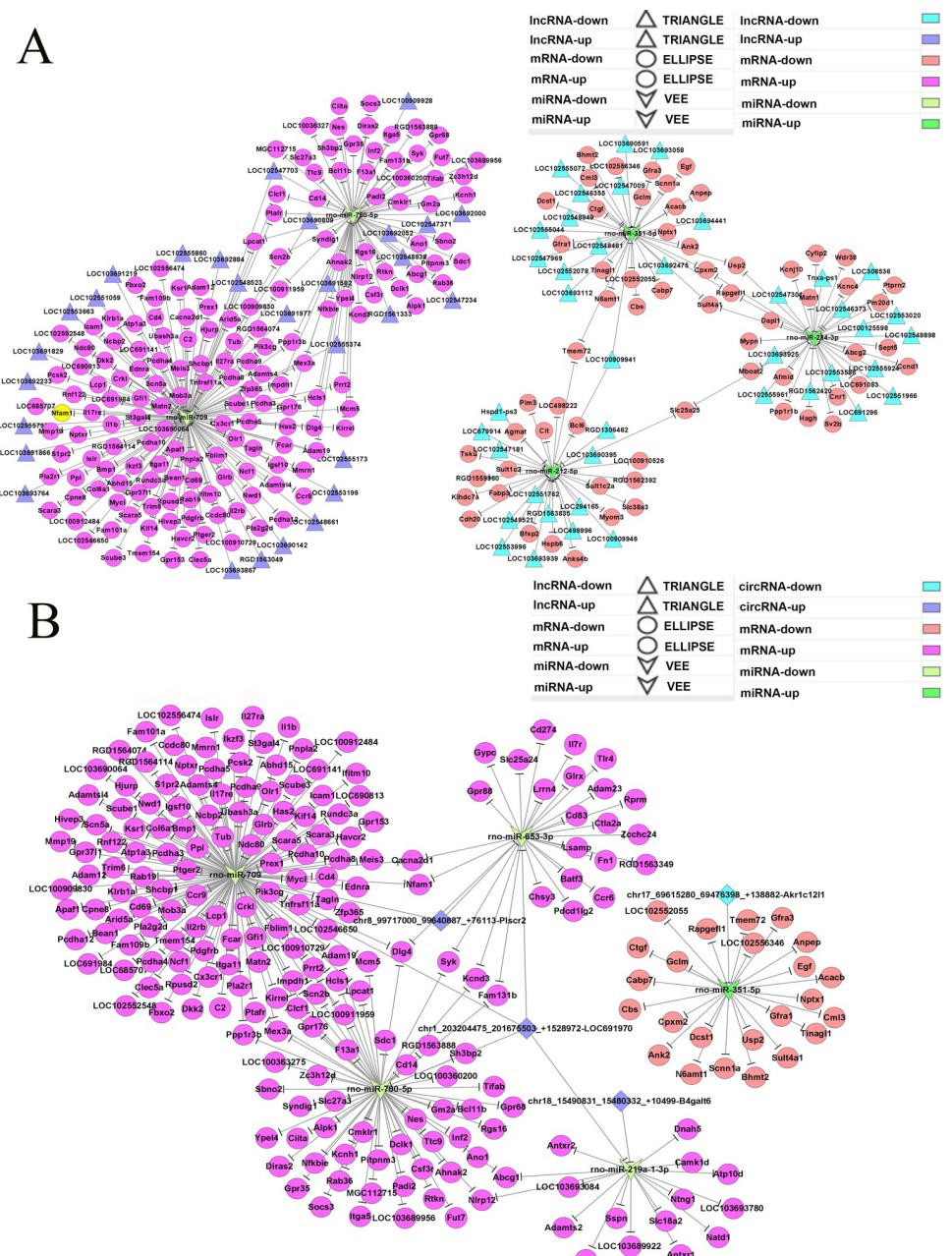

**Fig 7. The competing endogenous RNA network for the 5 miRNAs with maximum interactions.** (A) lncRNA-miRNA-mRNA competing endogenous RNA network. Multiple colours represent upregulation and downregulation of miRNAs, lncRNAs and mRNAs in the case sample compared with the control. Vee, triangle, and ellipse indicate DE-miRNAs, lncRNAs and mRNAs, respectively. (B) The circRNA-miRNA-mRNA competing endogenous RNA network. Multiple colours represent upregulation and downregulation of miRNAs, circRNAs and mRNAs in the case sample compared with the control. Vee, diamond, and ellipse indicate DE-miRNAs, circRNAs and mRNAs, respectively.

pathways that might be involved in the constructed lncRNA-miRNA-mRNA network in HN, we conducted GO functional enrichment analysis. The mRNA in ceRNA analysis was taken as the research target, and the significant GO class and its associated genes were obtained through gene function analysis (Fig 8A). Their BPs were involved in cell adhesion, inflammatory

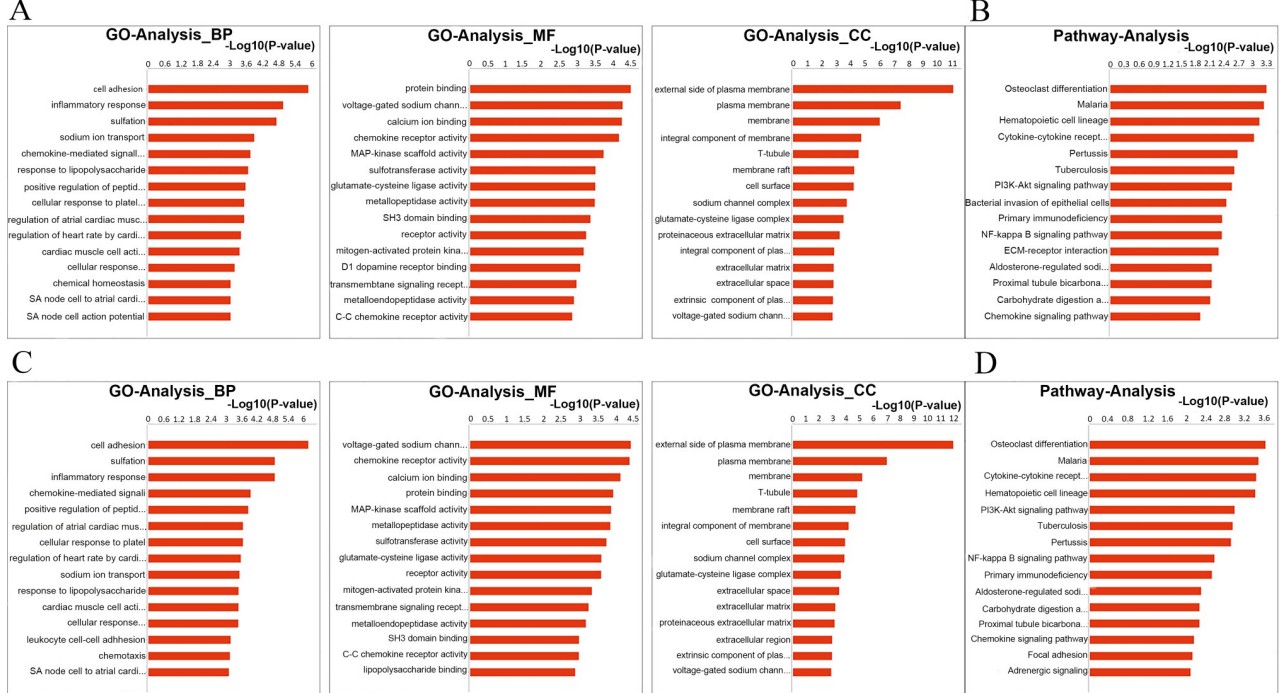

**Fig 8. GO and KEGG analysis involving DE-mRNAs in the lncRNA/circRNA-miRNA-mRNA competing endogenous RNA network.** (A) The top 30 GO terms of DE-mRNAs in the lncRNA-miRNA-mRNA competing endogenous RNA network, including MF, CC and BP classification. The horizontal axis is the enrichment value of functional degree, and the vertical axis is the entry name corresponding to GO in the Gene Ontology database. (B) KEGG analysis of DE-mRNAs in the lncRNA-miRNA-mRNA competing endogenous RNA network. (C) The top 30 GO terms of DE-mRNAs in the circRNA-miRNA-mRNA competing endogenous RNA network, including MF, CC and BP classification. The horizontal axis is the enrichment value of functional degree, and the vertical axis is the entry name corresponding to GO in the Gene Ontology database. (D) KEGG analysis ofDE-mRNAs in the circRNA-miRNA-mRNA competing endogenous RNA network. GO, Gene Ontology; KEGG, Kyoto Encyclopedia of Genes and Genomes pathway; MF, molecular function; CC, cellular component; BP, biological process.

response, sulfation, etc. KEGG pathway enrichment analysis was conducted (Fig 8B). The top 5 pathways revealed by KEGG pathway enrichment analysis were mainly involved in proteoglycans, osteoclast differentiation, malaria, haematopoietic cell lineage, cytokine-cytokine receptor interaction and pertussis.

## The circRNA-miRNA-mRNA ceRNA network map and analysis

The circRNA-miRNA-mRNA ceRNA network map consisted of 392 ceRNAs (S9 Table). Simultaneously, we focused on the top 5 DE-miRNAs with the most interactions and used these interactions to compose the ceRNA network (consisting of 1 upregulated miRNA, rno-miR-351-5p and 4 downregulated miRNAs, rno-miR-709, rno-miR-760-5p, rno-miR-653-3p, rno-miR-219a-1-3p) (Fig 7B). Examples included Cacna2d1/Nfam (up)-rno-miR-709/rno-miR-653-3p (down)-chr8_99717000_99640887_+76113-Plscr2 (up), Dlg4/Nfam1 (up)- rno-miR-709/rno-miR-653-3p (down)-chr1_203204475_201675503_+1528972_LOC691970 (up) and Cpxm2 (down)-rno-351-475p_203204475_201675503_+15972_LOC691970 (up) and Cbs/Cbs/Cpxm2 (down)-miR-8691c. Similarly, we performed GO analysis of DE-mRNAs involved in the ceRNA network (Fig 8C). CC analysis indicated that the proteins encoded by those DE-mRNAs are mainly located on the external side of the plasma membrane, plasma membrane, membrane, etc. Their MFs are involved in voltage-gated sodium channel activity, cardiac muscle cell action potential, chemokine receptor activity, and calcium ion binding, which mainly participate in BPs such as cell adhesion, sulfation, and the inflammatory

response. KEGG pathway enrichment analysis was conducted (Fig 8D). The top 5 pathways revealed by KEGG pathway enrichment analysis were mainly involved in osteoclast differentiation, malaria, cytokine-cytokine receptor interaction, haematopoietic cell lineage, and the PI3K-Akt signalling pathway.

## Validation of representative lncRNAs, miRNAs and mRNAs

To verify the RNA-seq results, four mRNAs, two lncRNAs and two miRNAs that emerged in the ceRNA network were chosen and verified by real-time quantitative polymerase chain reaction (RT-qPCR). The results showed that Adam19 and A2m were upregulated and the expression of Nlrp12, Ptafr did not differ significantly. Results for the miRNAs showed that the expression of miR-351-5p was upregulated in the renal samples of HN group rats compared to the control rats; in contrast, miR-760-5p was downregulated. Meanwhile, the expression of LNC102555374 and LNC102547703 was upregulated in kidney samples of HN group rats compared to the control rats (Fig 9). Thus, the RT-qPCR results revealed that the relative expression levels of the selected gene and ncRNAs were consistent with the RNA-seq data, which demonstrates the reliability of the RNA-seq results.

## Discussion

Along with dramatic changes in fundamental lifestyles, the prevalence of HUA has continuously increased in recent decades, such that HUA has become a fundamental health problem in industrialized nations [47]. Additionally, there is a gradual trend of younger HUA patients. HN is regarded as a common clinical complication of HUA, which threatens human health worldwide. Therefore, studying the mechanisms involved in HN is essential to prevent renal impairment. At present, the understanding of gene regulation in HN is still limited. Along with the advancement of molecular biotechnology, ncRNAs have recently attracted increasing attention [32]. RNA-seq technique was employed to attempt a preliminary indication of the

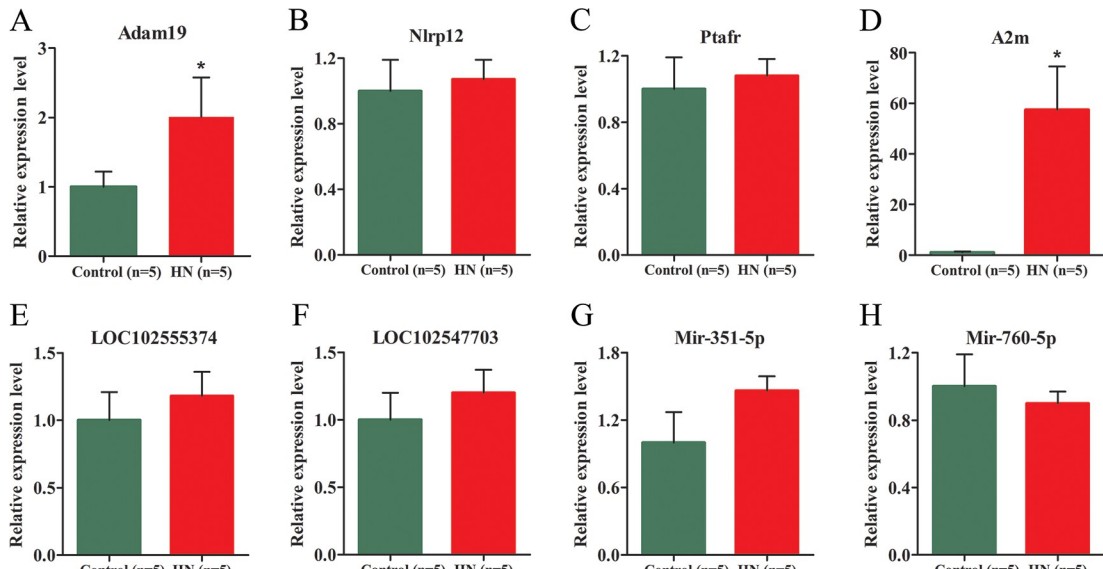

**Fig 9. qRT-PCR validation of representative mRNAs, lncRNAs and miRNAs.** The relative expression levels of mRNAs (A-D), lncRNAs (E, F) and miRNAs (G, H) were validated in kidney samples from HN rats (n = 5) and control rats by qRT-PCR for comparison to the RNA-seq results. For the comparison of quantitative data between groups, Student's t-test was used. Differences in means were considered statistically significant at *P < 0.05.

DE-transcripts and key signaling pathways associated with HN. GO and KEGG enrichment analysis results were executed to analyse the DEGs. Intriguingly, ceRNA networks were constructed to further discover the pathogenesis of HN.

In our research, we identified 2631 DE-transcripts in total, including 2250 mRNAs, 70 miR-NAs, 306 lncRNAs and 5 circRNAs. The top 5 upregulated genes included Mmp7, Cxcl6, Tdo2, Chst5, and Nos2, while the top 5 downregulated genes included Klk1c6, Slc22a13, Nhlh2, Cyp3a71-ps, and Scgb1c. Mmp7 is a secreted zinc-dependent endopeptidase involved in the regulation of kidney homeostasis [48]. MMP-7 is barely expressed in healthy adult kidneys [49], but both animal models and clinical findings suggest that its expression is upregulated in chronic kidney disease [50, 51]. Thus, MMP-7 has great potential as a clinical biomarker and therapeutic target for CKD. SLC22A13 has so far been determined to be a urate transporter by in vitro analysis, and studies have uncovered that dysfunctional variant of SLC22A13 reduce both gout risk and serum uric acid levels, suggesting that SLC22A13 is physiologically implicated in uric acid reabsorption in the human kidney [52]. Likewise, CXCL6 is a potential novel therapeutic target and candidate biomarker for JAK/STAT3 signaling in the treatment of diabetic nephropathy. Some findings indicate that TDO2 may play an important role in kidney disease progression and may be a promising marker for targeted therapy in renal cell carcinoma [53]. The top 5 upregulated lncRNAs included LOC102555798, LOC102550121, LOC103692116, LOC103692133 and LOC103693273, while the top 5 downregulated lncRNAs included LOC102554996, Slc22a7-ps1, RGD1560703, LOC685876, and LOC102547378. Our results showed that the expression pattern of ncRNAs in kidney tissue of HN was significantly different from that in the control group, which suggests that ncRNAs may play a vital role in the pathogenesis of HN and may become a potential molecular biomarker for HN. GO analysis revealed 2250 DE-mRNAs whose expression correlated with nephropathy and were associated with 831 BPs, 104 CCs, and 197 MFs. Genes that we identified to be involved in HN affected general BPs in the kidney, such as the inflammatory response, innate immune response, cell adhesion, chemotaxis and response to lipopolysaccharide. And we fund that HN is associated with 90 KEGG pathways. Chemokine signalling, TNF signalling, NOD-like receptor signalling and NF-κB signalling pathways were enriched significantly, which lead to renal hypertrophy, fibrosis and inflammation [54–57]. TNF is an important cytokine that induces multiple intracellular signalling pathways, such as apoptosis, cell survival, inflammation and immunity, and activated TNF assembles into a homotrimer that binds to its receptors (TNFR1, TNFR2), leading to trimerization of TNFR1 or TNFR2. TNFR1 signalling induces the activation of many genes, which are mainly controlled by two different pathways, namely, the NF-κB pathway and the MAPK cascade, or apoptosis and necrotizing ptosis. TNFR2 signalling activates NF-κB pathways, including PI3K-dependent NF-κB and JNK pathways. TNF is involved in the pathogenesis of many kidney diseases, including ischaemic kidney injury, renal graft rejection, and glomerulonephritis, which is often part of systemic vasculitis [57]. Studies have also revealed that the TNF pathway plays an essential part in the progression of diabetic nephropathy [58]. Analysis of the lncRNA-miRNA-mRNA ceRNA and the circRNA-miRNA-mRNA ceRNA networks showed an enrichment in PI3K-Akt signalling, NF-κB signalling and chemokine signalling pathways. The PI3K Akt pathway is an intracellular signal transduction pathway that promotes metabolism, proliferation, cell survival, growth and angiogenesis in response to extracellular signals [59]. Atractylenolide III was reported to attenuate muscle wasting in CKD via oxidative stress-mediated PI3K/AKT/mTOR pathway [60]. Previous studies have suggested that protease-activated receptor-2 can inhibit autophagy through the PI3K/Akt/mTOR signalling pathway, thus promoting renal tubular epithelial inflammation [61]. From the KEGG analysis, NF-κB signalling pathway was also of extreme interest to us because it is widely acknowledged as a typical pro-inflammatory

signalling pathway based on the activation of NF-κB by pro-inflammatory cytokines such as interleukin-1 (IL-1) and tumour necrosis factor A (TNF-α), as well as NF-κB-activated expression of other pro-inflammatory genes (including cytokines and chemokines) and adhesion molecules [57]. It has previously been reported that uric acid inhibits the proliferation of renal proximal tubular cells via NF-κB [62]. Therefore, we speculate that the mechanism of HN may be related to abnormal regulation of the PI3K-Akt signalling pathway and NF-κB signalling pathway.

In the ceRNA network analysis, some genes attracted our attention. PTAFR is a protein-coding gene that encodes seven transmembrane G-protein coupled receptors for platelet activation factor (PAF) located in the lipid raft and fossa of the cell membrane. PAF is a phospholipid that plays an important role in tumour transformation, tumour growth, angiogenesis, metastasis and pro-inflammatory processes [63]. And it was shown that PTAFR is one of the central genes of 18β-glycyrrhetinic acid to alleviate renal fibrosis by inhibiting the inflammatory response [64]. Additionally, NLRP12 encodes a member of the Caterpillar family of cytoplasmic proteins, which has been shown to play a significant role in the formation of inflammation against specific infections and act as a regulator of inflammatory signals [10]. ADAM19 encodes a member of the ADAM family, which is a type I transmembrane protein and is a marker of dendritic cell differentiation. It has been demonstrated to be an active metalloproteinase that may be involved in cell-cell and cell-matrix interactions and TNF-α shedding. It is proposed to play a role in pathological processes, such as cancer, inflammatory diseases and renal diseases. Abnormally high expression of ADAM19 is also linked to inflammation and fibrosis of the kidney. The targeted inhibition of ADAM19 may be crucial for the treatment of certain types of tumours and inflammatory diseases. An abnormally high expression of ADAM19 is also associated with nephritis and fibrosis, while the niche targeting inhibition of ADAM19 may be essential for the treatment of inflammatory diseases [65, 66]. Thus, we guess that PTAFR, NLRP12 and ADAM19 are closely related to the inflammatory response, which suggests that these genes may be related to the HN mechanism.

We compared the expression profiles of kidney samples from rats with HN with those from healthy rats by sequencing the entire transcriptome. We screened genes and signalling pathways associated with HN to explain the possible mechanism through bioinformatics analysis. While these studies reveal some important findings, there are limitations. The study was limited by its small sample size; however, the accuracy of the RNA-seq analysis was verified by qRT-PCR data, which indicated that the RNA-seq data were reliable. In future studies we will adopt immunoprecipitation to further explore the transcriptional regulatory network [28]. In summary, we identified some degree of HN at the genetic level. Changes in gene expression lead to changes in gene balance and signalling pathway expression. These discoveries advance our understanding of the HN mechanism, providing novel targets and a theoretical basis for the treatment of HN disease.

## Supporting information

**S1 Table. Primer sequences used for quantitative real-time polymerase chain reaction expression analysis.**
(DOCX)

**S2 Table. More detail information about the differentially expressed mRNA in the hyperuricaemic nephropathy group.**
(XLSX)

**S3 Table. List of all Gene Ontology term of the differentially expressed mRNA.**
(ZIP)

**S4 Table. List of all pathway terms of the differentially expressed mRNA.**
(XLSX)

**S5 Table. More detail information about the differentially expressed lncRNA in the hyperuricaemic nephropathy group.**
(XLSX)

**S6 Table. More detail information about the differentially expressed circRNA in the hyperuricaemic nephropathy group.**
(XLS)

**S7 Table. More detail information about the differentially expressed miRNA in the hyperuricaemic nephropathy group.**
(XLSX)

**S8 Table. All transcripts involved in lncRNA-miRNA-mRNA ceRNA network.**
(XLSX)

**S9 Table. All transcripts involved in cirRNA-miRNA-mRNA ceRNA network.**
(XLSX)

## Author Contributions

**Conceptualization:** Hengxiu Yan, Xiaoni Shao.

**Data curation:** Na Li, Mukaram Amatjan, Pengke He.

**Formal analysis:** Meiwei Wu.

**Funding acquisition:** Xiaoni Shao.

**Investigation:** Na Li, Pengke He.

**Methodology:** Na Li, Mukaram Amatjan, Pengke He, Meiwei Wu, Hengxiu Yan, Xiaoni Shao.

**Project administration:** Xiaoni Shao.

**Resources:** Xiaoni Shao.

**Software:** Na Li, Mukaram Amatjan, Pengke He.

**Supervision:** Hengxiu Yan, Xiaoni Shao.

**Validation:** Meiwei Wu.

**Visualization:** Na Li, Mukaram Amatjan.

**Writing – original draft:** Na Li, Mukaram Amatjan, Pengke He.

**Writing – review & editing:** Xiaoni Shao.

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
