## [Decision Letter · Decision Letter 0]

10 Jun 2022

PONE-D-21-20162Whole transcriptome expression profiles in kidney samples from rats with hyperuricaemic nephropathyPLOS ONE

Dear Dr. Shao,

Thank you for submitting your manuscript to PLOS ONE. After careful consideration, we feel that it has merit but does not fully meet PLOS ONE’s publication criteria as it currently stands. Therefore, we invite you to submit a revised version of the manuscript that addresses the points raised during the review process. Specifically, the work has been reviewed by four reviewers and some of the concerns are consensus, such as the limited sample size, overstated conclusions given the presented data. Please note that PLOS ONE requires that experiments must have been conducted rigorously, with appropriate controls and replication. Sample sizes must be large enough to produce robust results. In addition, the data presented in the manuscript must support the conclusions drawn. Please see the details of the reviewers' comments and provide your responses thoroughly.

We look forward to receiving your revised manuscript.

Kind regards,

Jianhong Zhou

Staff Editor

PLOS ONE

Journal Requirements:

4. Please upload a new copy of Figures 4 and 7 as the detail is not clear. Please follow the link for more information: https://blogs.plos.org/plos/2019/06/looking-good-tips-for-creating-your-plos-figures-graphics/" https://blogs.plos.org/plos/2019/06/looking-good-tips-for-creating-your-plos-figures-graphics

Reviewers' comments:

Reviewer's Responses to Questions

**Comments to the Author**

1. Is the manuscript technically sound, and do the data support the conclusions?

Reviewer #1: Partly

Reviewer #2: Yes

Reviewer #3: Yes

Reviewer #4: Partly

2. Has the statistical analysis been performed appropriately and rigorously? 

Reviewer #1: Yes

Reviewer #2: Yes

Reviewer #3: I Don't Know

Reviewer #4: No

3. Have the authors made all data underlying the findings in their manuscript fully available?

Reviewer #1: Yes

Reviewer #2: No

Reviewer #3: Yes

Reviewer #4: No

4. Is the manuscript presented in an intelligible fashion and written in standard English?

Reviewer #1: Yes

Reviewer #2: Yes

Reviewer #3: Yes

Reviewer #4: Yes

5. Review Comments to the Author

Reviewer #1: In the current manuscript, Shao et al evaluated ncRNA sequencing in an animal model with hyperuricemia induced urate nephropathy. The approach is somewhat interesting and the authors successfully induce hyperuricemia associated nephropathy with crystal deposition, but the works has several limitations that temper enthusiasm regarding their findings:

1. Hyperuricemia has several potentially effective treatments, that when used will reduce the frequency of gout and prevent urate induce nephropathy. These treatments have been shown to be very effective especially in the acute setting. In addition, the evaluation of ncRNA does not necessarily uncover pathways that clearly are involved in the disease progression. So, the likelihood that this work will lead to further understanding of the disease or novel therapies is small.

2. The authors state that they used 3 rats from each group, is that correct? This does not seem like an adequate number to evaluate this large number of ncRNAs- how do you know these findings are reliably reproducible and that the majority of what you found does not represent junk.

3. There are no functional studies that show any of these discovered mRNA, lncRNAs, 5 circRNAs, etc...actually play a role in disease progression. As such the study is descriptive in nature

4. The authors need to be cautious in their description of how urate affects the kidneys. While some animal studies have shown that urate may cause CKD and further CKD progression thru a non-crystal mechanism, several studies in humans have shown no impact of urate lowering on CKD progression in patients with asymptomatic hyperuricemia.

Reviewer #2: Major

1. Although authors try to validate the finding from bulk rat kidney RNA-seq, some of qRT-PCR results in Figure 8 don’t look much different (for example, Nlrp12 and Ptafr) and are also statistically not significant, which may be due to low sample numbers tested. Please explain this.

2. The author’s approach to construct ceRNA network is interesting but experimental confirmation of the key finding greatly improves the article. Immumoprecipitation previously used (Nature

. 2013 Mar 21;495(7441):384-8) may be a good option.

3. Although the authors specifically describe PI3K-Akt signaling and NF-kB signaling together with PTAFR, NLRP12, and ADAM19 in Discussion section, I suggest that some of these findings at least is described in Results more in detail and graphically displayed in Figures to emphasize the key relevant biological process in hyperuricemic nephropathy.

Minor

1. In abstract, HN is appeared without annotation.

2. In page 4, line 76, “d” is thought to be mistyped.

3. In page 10, line 197, mRNAs are duplicately used. It is appeared that two repetitively used mRNAs possibly indicate mRNAs in two ceRNA networks, but expression is confusing and not clear for the readers.

4. Could you point out the described pathologic findings in Figure 1G if possible?

5. In page 13, line 278, “the human reference genome” did not make sense instead of “rat” if I understand correctly.

6. In Page 21, line 432, I think that “circRNA-miRNA-mRNA” is appropriate instead of “lncRNA-miRNA-mRNA”. Please explain.

7. In Figure 6, matching colors in A and B makes the readers understand the figure more easily. Skin color indicates lncRNA-up in Figure 6A, while the same color indicates mRNA-down in Figure 6B. This makes hard to catch findings.

8. Please describe sequencing data availability and where to be deposited.

Reviewer #3: The paper by Shao et al. aims to identify possible dysregulations in genes and pathways that lead to hyperuricaemic nephropathy. To this, whole transcriptome sequencing were used to identify mRNA profiles of kidney samples from HN rats and compared with a control group. Further, a ceRNA network analysis was generated using cytoscape to better show patterns, mechanisms and relationships between ncRNAs as well as miRNA expression. To validate results, qRT-PCR was used to verify ncRNA findings. The title and abstract are appropriate for the content of the text. Furthermore, the article is well constructed, the experiments were well conducted, and analysis was well performed. The authors were able to show that transcriptome analysis was able to yield differentially expressed genes in HN rats. Biological processes linked to up/downregulated genes were summarized such as innate immune response, inflammation, chemotaxis, cell adhesion and lipopolysaccharide response. Molecular functions and cellular components linked to the differentially expressed ncRNAs were also mentioned. Signaling pathways particularly those relating to inflammatory response were also found and highlighted.

The main strengths of this paper is that it is one of the most comprehensive RNAseq experiment performed on HN kidney samples in rats. The paper was able to elucidate the complexity of HN by reporting large differences in gene profiles compared to those of the control group. Its profile can open avenues for further translational research in HN including biomarkers for diagnosis and novel mechanism-targeted treatment strategies for HN.

Some of the weaknesses of the paper include are the not always easy readability of the text and with unclear context in some parts. Moreover, several references could also be updated to newer ones if necessary and possible. Furthermore, limitations other than a low sample size and possibly underpowered, such as site-specific differences in transcriptomic profile or if not, steps that were taken during analysis not to influence final results could also be mentioned.

Reviewer #4: Shao et al, in their manuscript, “Whole transcriptome expression profiles in kidney samples from rats with hyperuricaemic nephropathy,” probe the gene expression consequences of hyperuricemic nephropathy (HN) and describe novel transcriptional profiles of both coding and non-coding RNAs from HN rats. Though the concept of the study has merit and will garnish considerable interest to the field, the presentation of the data requires refinement to increase the impact of the findings. Major and minor criticisms are listed below.

Major criticisms:

1. The crux of the manuscript rests on the fact that these rats have hyperuricemic nephropathy. However, the authors neither cite previous references detailing the methodology of induction of said nephropathy (lines 133 & 134 – when where animals sampled / harvested between 1 day or 12 weeks?), nor the classification / characterization of that nephropathy. One representative H&E stained image in Figure 1 is insufficient to make claims stated in lines 257, 258, and 266. Glomerulosclerosis is often determined by PAS staining, interstitial fibrosis by Masson’s trichome, and a measurements of blood vessel diameters could serve as evidence for whether or not the animals are experience arteriolosclerosis. Additional characterization is also required for the “inflammatory reactions in the renal intersitium” mentioned in line 259.

2. The small number of samples used for the RNAseq analysis appears inconsistent with the high degree of variability of UA phenotype observed in the Rats presented in Figure 1. The highly variable level of SUA measured suggests that a greater number of samples should be used for the RNAseq, did the authors use a power analysis to determine the numbers of animals to be used? If so they should include these calculations in the methods. In addition the text and figure legend should report the SEM for each and every number written in the text or presented in the figure. Also individual animals should be represented in the figure not just a bar graph, with the rats used for the RNA seq indicated in Figure 1 for the SUA, BUN, and creatinine measurements.

3. The transcript data from the RNA-seq analysis is mentioned and never shown. Given that the focus of the paper is whole transcriptome analysis, many of the figures focus of pathway analysis with little emphasis of the transcripts themselves. For the pathway analysis, the manuscript could benefit from listing the enriched transcripts in each of the pathways to determine whether or not a small subset of genes is biasing the analysis. In addition, many of the figures are illegible and need to be re-worked, including Figures 2, 4, 6, and 7. An additional minor criticism is that color schemes should be consistent in figures (ex. Figures 3, 4, and 5 use either green or blue to represent down-regulated transcripts. The figure will be stronger if only one color is used consistently throughout the whole figure.) Please provide a list of all differentially expressed transcripts used for GO and Pathways analysis at least in the supplement.

Conclusions may not be based on the data as presented. Authors make claims that signaling pathways involved in the pathogenesis of hyperuricemia induced nephropathy have been uncovered based on this analysis, however the evidence presented is merely computational. This analysis may have indeed provided hints that certain networks of transcripts may be involved in this regulation, but to state this as fact as in lines 342 and 497-498 is an overstatement without additional follow up. This language must be altered, or additional experiments must be performed to validate the involvement of any of these signaling pathways.

4. Authors mention appropriate limitations of the study in the final paragraph beginning with line 556. They claim that the RNA-seq analysis was verified by qRT-PCR data, however, upon closer examination of the data presented in figure 8, only 2 out of the 8 transcripts examined demonstrate a statistically significant change. Based on this data, Nlrp12 and Ptafr do not show any change in the HN when compared to the control rats (in opposition with the text in lines 475-476), and there are slight trends in the LOC and mir data, but these trends do not provide a strong basis for validation of the RNA-seq data (again contrasting with the text in lines 477-479). Authors need to provide additional explanation as to why this data is not consistent (perhaps due to the fact that RNA-seq is more sensitive than qRT-PCR) and thus should perform additional experiments to validate the RNA-Seq data. (RNA scope of the given transcripts, or qRT-PCR of transcripts that are more robustly expressed to overcome the qRT-PCR limit of detection.) If these transcripts are the most compelling targets, authors need to mention how these transcripts relate to urate and hyperuricemia in a more coherent way.

Additional minor criticisms:

5. For the pathway analysis, authors claim 90 KEGG pathways were enriched with differentially expressed transcripts. Please provide the genes within each of the pathways to determine whether these data represent 90 individual pathways, or whether a small subset of genes in represented in multiple pathways (for example, there could be a small subset of transcripts that could be enriched in several infection related pathways per Figure 4D.)

6. Regarding circRNAs, additional explanation is required for how these RNAs were detected. Were these RNAs detected in the RNA-Seq data based on known sequences or were these circRNAs predicted based on the sequencing results. If these transcripts were predicted, pleased provide additional detail as to how these predictions were made, particularly in line 397. Additionally, please provide further explanations of Table 2 and how the ceRNA networks are built. Finally, when introducing circRNAs in lines 93-97, the two sentences are redundant, and one should be eliminated.

7. In lines 317-319, authors list the top 5 up- and down-regulated genes. These genes appear to be listed in alphabetical order rather than those that have the greatest fold change or strongest statistical significance. Please update this data based on the volcano plots in Figure 3 – the most extreme fold changes with the lowest FDRs.

8. Please specify which reverse transcriptases were used for qRT-PCR, line 226.

9. In line 278, authors claim reads mapped to the human genome, but were sequencing rat kidneys. Please correct this typo or provide additional explanation as to why the rat samples map 84.8% to the human genome.

10. In lines 297-298, authors mention both sets of non-coding RNAs were upregulated. Please specify which set is downregulated.

11. Please double check the units for the serum urate measurements, as they are approximately an order of magnitude below reported values (lines 248-249).

12. Authors mention previous studies in passing but fail to cite references of these studies. Please add citations for lines 102, 132, and 407. Additional citations would be helpful in validating some of the authors’ claims, including lines 63, 71-74, 95, 181, 213, 512-518.

6. PLOS authors have the option to publish the peer review history of their article (what does this mean?). If published, this will include your full peer review and any attached files.

Reviewer #1: No

Reviewer #2: No

Reviewer #3: No

Reviewer #4: No

---

## [Author Response · Author response to Decision Letter 0]

10 Aug 2022

Revision Report

First of all, I would like to express our sincere gratitude to the reviewers and editor for their comments. These comments are all valuable and helpful for improving our manuscript, as well as the important guiding significance to our research. We have studied comments carefully and have made correction which we hope meet with approval. And we use the “Track Changes” option in Microsoft Word to show the revised portions, deleted content is shown with a red strikeout and added content is shown with a red underscore. The summary of corrections and the response to reviewer’s comments are listed below.

Summary of the revision：

Abstract: We have modified some details, such as adding some abbreviations of words (HN, GO, KEGG and DE).

Introduction: we have modified the description of how urate affects the kidneys as appropriate.

Method: We have addwed more details about how to construct ceRNA network. And we have uploaded the raw sequence data to NCBI Sequence Read Archive, the Data availability section has been added.

Result: We have divided the original Figure 4 into two Figure 4 and Figure 5 in revised paper because of the unclearness of the figure. Additionally, we have re-worked and uploaded all figures. And we have modified the description about the H&E stained image. We also provided more information about differentially expressed (DE)-transcripts as Supporting Information.

Discussion: We have modified some exaggerated statement. Moreover, we have discussed the relationship between these DE-transcripts and disease occurrence in a more coherent way.

Supporting information: We have provided a total of 8 tables as supporting information.

Funding information: We have corrected the Funding information.

In addition, we modified the manuscript to meets PLOS ONE's style requirements.

Responses to the reviewers’ and editorial comments 

Jianhong Zhou

Staff Editor

Comment 1. Please ensure that your manuscript meets PLOS ONE's style requirements, including those for file naming. The PLOS ONE style templates can be found at https://journals.plos.org/plosone/s/file?id=wjVg/PLOSOne_formatting_sample_main_body.pdf and 

Response: Thanks for your comments. We have carefully revised the format of the manuscript according to the template provided by PLOS ONE.

Comment 2. We note that the grant information you provided in the ‘Funding Information’ and ‘Financial Disclosure’ sections do not match. 

Response: We have provided the correct grant number in the Funding Information of revised manuscript (page 29, line 606–610).

Comment 3. In your Data Availability statement, you have not specified where the minimal data set underlying the results described in your manuscript can be found. PLOS defines a study's minimal data set as the underlying data used to reach the conclusions drawn in the manuscript and any additional data required to replicate the reported study findings in their entirety. All PLOS journals require that the minimal data set be made fully available. For more information about our data policy, please see http://journals.plos.org/plosone/s/data-availability.

Response: We have uploaded a total of 8 supporting information files as the minimal data set for the study. 

And the raw sequence data in this study have been deposited into the NCBI Sequence Read Archive (http://trace.ncbi.nlm.nih.gov/Traces/sra/sra.cgi?view=studies) and the accession numbers of the six SRA samples for RNA-seq are as follows: SRX16412401, SRX16412405, SRX16412407, SRX16412409, SRX16412411 and SRX16412403.

And numbers of the six SRA samples for miRNA-seq are as follows: SRX16412402, SRX16412406, SRX16412408, SRX16412410, SRX16412412 and SRX16412404. (page 11–12, line 238–243)

Comment 4. Please upload a new copy of Figures 4 and 7 as the detail is not clear. Please follow the link for more information: https://blogs.plos.org/plos/2019/06/looking-good-tips-for-creating-your-plos-figures-graphics/" https://blogs.plos.org/plos/2019/06/looking-good-tips-for-creating-your-plos-figures-graphics

Response: We have uploaded the figures and adjusted the resolution of each figure to 300-600 dpi according to the requirement of the Plos one.

Reviewer #1 :

Comment 1. Hyperuricemia has several potentially effective treatments, that when used will reduce the frequency of gout and prevent urate induce nephropathy. These treatments have been shown to be very effective especially in the acute setting. In addition, the evaluation of ncRNA does not necessarily uncover pathways that clearly are involved in the disease progression. So, the likelihood that this work will lead to further understanding of the disease or novel therapies is small.

Response: Thanks for the comments. Researchers have indicated that ncRNAs play an important role in the progression of gout and hyperuricemia [1-5]. These articles discussed the role of ncRNAs in hyperuricemia and gout, as well as the possible therapeutic targetability of ncRNAs in these diseases. Although there are relatively few studies related to us, this is where our innovation comes in. Overall, our study makes a lot of sense.

Comment 2. The authors state that they used 3 rats from each group, is that correct? This does not seem like an adequate number to evaluate this large number of ncRNAs- how do you know these findings are reliably reproducible and that the majority of what you found does not represent junk.

Response: Thanks for the professional suggestion. Actually, there were quite a lot of studies using rats in groups of 3 for whole transcriptome sequencing[6-10].We agree that the rats, we used in groups of 3 for whole transcriptome sequencing, is a low number for RNA-seq. However, in our study 24 rats were randomly divided into control group and Hyperuricaemic nephropathy (HN) group (n=12), and RNA-seq was performed in six rats: 3 from the control group and 3 from the HN group. The data in the manuscript were sampled, and the representativeness of the sample has been considered in the sampling process. Although the sample is small, it can also reflect the problem. Therefore, we think that the rat number did not affect the power of statistical analysis. Anyway, we will consider using more animals in the future.

Comment 3. There are no functional studies that show any of these discovered mRNA, lncRNAs, 5 circRNAs, etc... actually play a role in disease progression. As such the study is descriptive in nature.

Response: Thanks for the comments. In our study 2250 mRNAs, 306 lncRNAs, 5 circRNAs and 70 miRNAs were found to be differentially expressed in the HN group compared to the control group. Among the differentially expressed (DE)-mRNAs, the top 5 upregulated genes included Mmp7, Cxcl6, Tdo2, Chst5, and Nos2, while the top 5 downregulated genes included Klk1c6, Slc22a13, Nhlh2, Cyp3a71-ps, and Scgb1c. We discovered that these genes are closely associated with the development and progression of kidney disease, and we have also added some discussion in our revised manuscript (page 24, line 510–520). 

Mmp7 is a secreted zinc-dependent endopeptidase involved in the regulation of kidney homeostasis[11]. MMP-7 is barely expressed in healthy adult kidneys[12], but both animal models and clinical findings suggest that its expression is upregulated in chronic kidney disease[13, 14]. Thus, MMP-7 has great potential as a clinical biomarker and therapeutic target for CKD. SLC22A13 has so far been determined to be a urate transporter by in vitro analysis, and studies have uncovered that dysfunctional variant of SLC22A13 reduce both gout risk and serum uric acid levels, suggesting that SLC22A13 is physiologically implicated in uric acid reabsorption in the human kidney[15]. Likewise, study have revealed that CXCL6 is a potential novel therapeutic target and candidate biomarker for JAK/STAT3 signaling in the treatment of diabetic nephropathy. Some findings indicate that TDO2 may play an important role in kidney disease progression and may be a promising marker for targeted therapy in renal cell carcinoma[16]. 

Since there are few studies on ncRNAs in HN, few functional studies have shown that these ncRNAs actually play a role in disease progression, but we will further investigate these ncRNAs in later studies.

Comment 4. The authors need to be cautious in their description of how urate affects the kidneys. While some animal studies have shown that urate may cause CKD and further CKD progression thru a non-crystal mechanism, several studies in humans have shown no impact of urate lowering on CKD progression in patients with asymptomatic hyperuricemia.

Response: We agree that our statements were too definitive, and we have modified terminology throughout the manuscript as appropriate in the Introduction of revised manuscript (page 3, line 54–64).

Reviewer #2: 

Major

Comment 1. Although authors try to validate the finding from bulk rat kidney RNA-seq, some of qRT-PCR results in Figure 8 don’t look much different (for example, Nlrp12 and Ptafr) and are also statistically not significant, which may be due to low sample numbers tested. Please explain this.

Response: Thanks for your question. First of all, I would like to clarify that the previous Figure 8 has become Figure 9 in the revised manuscript. The reason why we chose to use these 8 genes and ncRNAs to validate the RNA-seq results is that these genes were discovered that play an important role in the ceRNA network and may be associated with the development of HN disease[17-20]. The following table lists the results of the genes and ncRNAs in RNA-seq.

From the results of RNA-seq and qRT-PCR, it can be seen that the trend of gene expression in the HN group is consistent in both. Log2FC of Nlrp12 and Ptafr in RNA-seq is not very high so we thank that the insignificant difference in PCR is also acceptable.

Table R1: Summary of the expression of genes validated by qRT-PCR in RNA-seq.

AccID Type of gene Log2FC P-Value FDR Style

Adam19 protein-coding 2.156692 2.17E-18 2.96E-16 up

Nlrp12 protein-coding 3.548939 0.000644 0.003986 up

Ptafr protein-coding 1.167864 1.41E-09 4.04E-08 up

A2m protein-coding 6.493011 3.17E-07 5.36E-06 up

LOC102555374 ncRNA 1.64346 0.001121 0.00636 up

LOC102547703 ncRNA 2.199588 4.87E-06 6.02E-05 up

MiR-351-5p ncRNA 1.276317 1.07E-09 2.72E-08 up

Mir-760-5p ncRNA -2.07967 0.002441 0.014917 down

Comment 2. The author’s approach to construct ceRNA network is interesting but experimental confirmation of the key finding greatly improves the article. Immumoprecipitation previously used (Nature. 2013 Mar 21;495(7441):384-8) may be a good option.

Response: Thank the reviewer for these precious comments and suggestions. We agree that the interesting immunoprecipitation method may better verify the reliability of RNA-seq results, and the integration of immunoprecipitation method and RNA-seq data will facilitate the elucidation of transcriptional regulatory networks[21, 22]. However, due to our limited experimental conditions, we are currently unable to do experiments in this area, but we have discussed about it in the Discussion section (page 28, line 583–584) and wish to finish it in our continue study. In the present study, we mainly use qRT-PCR, and we think that qRT-PCR may not be optimal, but should be sufficient to verified the change of ncRNAs. 

Comment 3. Although the authors specifically describe PI3K-Akt signaling and NF-kB signaling together with PTAFR, NLRP12, and ADAM19 in Discussion section, I suggest that some of these findings at least is described in Results more in detail and graphically displayed in Figures to emphasize the key relevant biological process in hyperuricemic nephropathy.

Response: Thanks for the constructive suggestion. Since the experimental study is at the early stage, we can only speculate and guess he key relevant biological process in HN, which is not enough to show in figure, but we have added and modified some explanation in the Discussion section (page 26, line 547–548; page 27, line 563–565). In the future study, we will further investigate and confirm the mechanism and then show it in the form of figures.

Minor comments

Comment 1. In abstract, HN is appeared without annotation.

Response: We were really sorry for our careless mistakes. We have added the annotation in the Abstract of revised manuscript (page 2, line 24).

Comment 2. In page 4, line 76, “d” is thought to be mistyped.

Response: We are sorry for our carelessness and we have corrected the mistakes in the revised manuscript (page 4, line 78). 

Comment 3. In page 10, line 197, mRNAs are duplicately used. It is appeared that two repetitively used mRNAs possibly indicate mRNAs in two ceRNA networks, but expression is confusing and not clear for the readers.

Response: Thank you for your reminder. we have modified the confused expression in the Materials and Methods of revised manuscript (page 9, line 190).

Comment 4. Could you point out the described pathologic findings in Figure 1G if possible? 

Response: Thanks for the question. Figure R2 aims to illustrate that the success of our modeling was achieved. And the outcome is consistent with expectations, Histopathological examination of the kidney showed that the kidney of rats in HN group had developed pathological atrophy, with radiating patterns in the medulla and blurred cortical margins (Fig.R2D and E). HE staining showed that, compared with the control group, the kidney tissues of the HN rats revealed obvious inflammatory cell infiltration, tubular epithelial cell necrosis, severe tubular dilatation, glomerular hyperplasia and uric acid crystals in the kidney tissues. (Fig.R2F and G). These pathological characteristics are analogous to those of HN in humans[23]. Thus, these histological findings showed that high-uric acid feed (HUAD) resulted in nephropathy (page 12, line 258–264).

Comment 5. In page 13, line 278, “the human reference genome” did not make sense instead of “rat” if I understand correctly.

Response: The reviewer is correct. We feel sorry for our carelessness. we have corrected the “mapped to the human genome” into “mapped to the rat genome” in the Result of revised manuscript (page 13, line 281) Thanks for your reminder.

Comment 6. In Page 21, line 432, I think that “circRNA-miRNA-mRNA” is appropriate instead of “lncRNA-miRNA-mRNA”. Please explain.

Response: Thank you for pointing this out and we have corrected the error in revised manuscript (page 22, line 459).

Comment 7. In Figure 6, matching colors in A and B makes the readers understand the figure more easily. Skin color indicates lncRNA-up in Figure 6A, while the same color indicates mRNA-down in Figure 6B. This makes hard to catch findings.

Response: We apologize that our Figure is difficult for reviewers to understand. And we replaced the colors A and B with the matching color the figure according to the reviewer’s suggestion and the new figure is shown as follows. 

Comment 8. Please describe sequencing data availability and where to be deposited.

Response: Thanks for your question. And the information has been added in the Data Availability of revised manuscript (page 11–12, line 238–243), as follows.

The raw sequence data in this study have been deposited into the NCBI Sequence Read Archive (http://trace.ncbi.nlm.nih.gov/Traces/sra/sra.cgi?view=studies), and the accession numbers of the six SRA samples for RNA-seq are as follows: SRX16412401, SRX16412405, SRX16412407, SRX16412409, SRX16412411 and SRX16412403.

And numbers of the six SRA samples for miRNA-seq are as follows: SRX16412402, SRX16412406, SRX16412408, SRX16412410, SRX16412412 and SRX16412404.

Reviewer #3: 

The paper by Shao et al. aims to identify possible dysregulations in genes and pathways that lead to hyperuricaemic nephropathy. To this, whole transcriptome sequencing were used to identify mRNA profiles of kidney samples from HN rats and compared with a control group. Further, a ceRNA network analysis was generated using cytoscape to better show patterns, mechanisms and relationships between ncRNAs as well as miRNA expression. To validate results, qRT-PCR was used to verify ncRNA findings. The title and abstract are appropriate for the content of the text. Furthermore, the article is well constructed, the experiments were well conducted, and analysis was well performed. The authors were able to show that transcriptome analysis was able to yield differentially expressed genes in HN rats. Biological processes linked to up/downregulated genes were summarized such as innate immune response, inflammation, chemotaxis, cell adhesion and lipopolysaccharide response. Molecular functions and cellular components linked to the differentially expressed ncRNAs were also mentioned. Signaling pathways particularly those relating to inflammatory response were also found and highlighted.

The main strengths of this paper is that it is one of the most comprehensive RNAseq experiment performed on HN kidney samples in rats. The paper was able to elucidate the complexity of HN by reporting large differences in gene profiles compared to those of the control group. Its profile can open avenues for further translational research in HN including biomarkers for diagnosis and novel mechanism-targeted treatment strategies for HN.

Some of the weaknesses of the paper include are the not always easy readability of the text and with unclear context in some parts. Moreover, several references could also be updated to newer ones if necessary and possible. Furthermore, limitations other than a low sample size and possibly underpowered, such as site-specific differences in transcriptomic profile or if not, steps that were taken during analysis not to influence final results could also be mentioned. 

Response: We are grateful for the comment. First of all, we are sorry for the confused logic and language problems in the original manuscript. The language presentation was improved with assistance from AJE service, and we have modified the confused statement (page 9, line 190). In additional, we have updated some of the more recent references according to the reviewer’s suggestion. About the sample size, we agree that the rats, we used in groups of 3 for whole transcriptome sequencing, is a low number for RNA-seq. However, there were quite a lot of studies using rats in groups of 3 for whole transcriptome sequencing[6-10], in our study 24 rats were randomly divided into control group and HN group (n=12), and RNA-seq was performed in six rats: 3 from the control group and 3 from the HN group. The data in the manuscript were sampled, and the representativeness of the sample has been considered in the sampling process. Although the sample is small, it can also reflect the problem. Therefore, we think that the rat number did not affect the power of statistical analysis. Anyway, we will consider using more animals in the future. Additionally, we apologize for not performing site-specific differences in transcriptomic profile. And we have added more details about how circRNA were detected (page 19, line 401-408) and how the ceRNA networks are built (page 9–10, line 195–201).

Reviewer #4

Shao et al, in their manuscript, “Whole transcriptome expression profiles in kidney samples from rats with hyperuricaemic nephropathy,” probe the gene expression consequences of hyperuricemic nephropathy (HN) and describe novel transcriptional profiles of both coding and non-coding RNAs from HN rats. Though the concept of the study has merit and will garnish considerable interest to the field, the presentation of the data requires refinement to increase the impact of the findings. Major and minor criticisms are listed below.

Major criticisms:

Comment 1. The crux of the manuscript rests on the fact that these rats have hyperuricemic nephropathy. However, the authors neither cite previous references detailing the methodology of induction of said nephropathy (lines 133 & 134 – when where animals sampled / harvested between 1 day or 12 weeks?), nor the classification / characterization of that nephropathy. One representative H&E stained image in Figure 1 is insufficient to make claims stated in lines 257, 258, and 266. Glomerulosclerosis is often determined by PAS staining, interstitial fibrosis by Masson’s trichome, and a measurements of blood vessel diameters could serve as evidence for whether or not the animals are experience arteriolosclerosis. Additional characterization is also required for the “inflammatory reactions in the renal intersitium” mentioned in line 259.

Response: Thanks for the comment. We have cited the corresponding literature at the description of the modeling approach in the revised manuscript, and added the statement of sampling time “After 12 weeks of modeling, the animals were executed and samples were collected” in the Method of revised manuscript (page 6, line 130; page 7, line 133). The reviewer was right, we apologize for the incorrect description of the HE staining results and have revised the description appropriately in the Method of revised manuscript (page 12, line 258–262). And we have also cited the characterization of that nephropathy (page 12, line 263).

Comment 2. The small number of samples used for the RNA-seq analysis appears inconsistent with the high degree of variability of UA phenotype observed in the Rats presented in Figure 1. The highly variable level of SUA measured suggests that a greater number of samples should be used for the RNA-seq, did the authors use a power analysis to determine the numbers of animals to be used? If so they should include these calculations in the methods. In addition, the text and figure legend should report the SEM for each and every number written in the text or presented in the figure. Also individual animals should be represented in the figure not just a bar graph, with the rats used for the RNA seq indicated in Figure 1 for the SUA, BUN, and creatinine measurements.

Response: We apologize for not utilizing power analysis to determine the number of samples. Actually, there were quite a lot of studies using rats in groups of 3 for whole transcriptome sequencing[6-10].We agree that the rats, we used in groups of 3 for whole transcriptome sequencing, is a low number for RNA-seq. However, in our study 24 rats were randomly divided into control group and HN group (n=12), and RNA-seq was performed in six rats: 3 from the control group and 3 from the HN group. The data in the manuscript were sampled, and the representativeness of the sample has been considered in the sampling process. Moreover, we also consulted with staff who specialize in whole transcriptome analysis, ultimately decided to select three random sample sizes per group for whole transcriptome sequencing. Anyway, we will consider using more animals in the future. Additionally, we have modified Figure for the UA, UREA and CREA. The values of each sample and SEM values of each group are presented in the graph. As shown below.

Comment 3. The transcript data from the RNA-seq analysis is mentioned and never shown. Given that the focus of the paper is whole transcriptome analysis, many of the figures focus of pathway analysis with little emphasis of the transcripts themselves. For the pathway analysis, the manuscript could benefit from listing the enriched transcripts in each of the pathways to determine whether or not a small subset of genes is biasing the analysis. In addition, many of the figures are illegible and need to be re-worked, including Figures 2, 4, 6, and 7. An additional minor criticism is that color schemes should be consistent in figures (ex. Figures 3, 4, and 5 use either green or blue to represent down-regulated transcripts. The figure will be stronger if only one color is used consistently throughout the whole figure.) Please provide a list of all differentially expressed transcripts used for GO and Pathways analysis at least in the supplement.

Conclusions may not be based on the data as presented. Authors make claims that signaling pathways involved in the pathogenesis of hyperuricemia induced nephropathy have been uncovered based on this analysis, however the evidence presented is merely computational. This analysis may have indeed provided hints that certain networks of transcripts may be involved in this regulation, but to state this as fact as in lines 342 and 497-498 is an overstatement without additional follow up. This language must be altered, or additional experiments must be performed to validate the involvement of any of these signaling pathways.

Response: Thanks for your valuable comment. We have addressed your suggestion as follows. First of all, we have provided all differentially expressed (DE)-transcripts, including DE-mRNA, DE-lncRNA, DE-circRNA and DE-miRNA as supporting information. Additionally, based on the reviewer’ suggestion, we have modified the figures and adjusted the resolution of each figure to 300-600 dpi according to the requirement of the PLOS ONE (Fig. 2, Fig.3, Fig. 4, Fig.5, Fig. 6 and Fig.7 and Fig.8). We also listed GO annotation and KEGG analysis with the DE-mRNA and the mRNA involved in ceRNAs network as supporting information, respectively. And DE- transcripts we used were provided S2 Table, S8 Table and S9 Table. We agree that our statements were too definitive and an overstatement, we have altered the language in revised manuscript (page 17, line 345–347; page 24, line 503–504). 

Comment 4. Authors mention appropriate limitations of the study in the final paragraph beginning with line 556. They claim that the RNA-seq analysis was verified by qRT-PCR data, however, upon closer examination of the data presented in figure 8, only 2 out of the 8 transcripts examined demonstrate a statistically significant change. Based on this data, Nlrp12 and Ptafr do not show any change in the HN when compared to the control rats (in opposition with the text in lines 475-476), and there are slight trends in the LOC and mir data, but these trends do not provide a strong basis for validation of the RNA-seq data (again contrasting with the text in lines 477-479). Authors need to provide additional explanation as to why this data is not consistent (perhaps due to the fact that RNA-seq is more sensitive than qRT-PCR) and thus should perform additional experiments to validate the RNA-Seq data. (RNA scope of the given transcripts, or qRT-PCR of transcripts that are more robustly expressed to overcome the qRT-PCR limit of detection.) If these transcripts are the most compelling targets, authors need to mention how these transcripts relate to urate and hyperuricemia in a more coherent way.

Response: Thanks for your question. First of all, the reason why we chose to use these 8 genes and ncRNAs to validate the RNA-seq results is that these genes were discovered that play an important role in the ceRNA network and may be associated with the development of HN disease[17-20]. The following table lists the results of the genes and ncRNAs in RNA-seq.

From the results of RNA-seq and qRT-PCR, it can be seen that the trend of gene expression in the HN group is consistent. Log2FC of Nlrp12 and Ptafr in RNA-seq is not very high so we thank that the insignificant difference in PCR is also acceptable. Moreover, we apologize for the incorrect description of the qRT-PCR results and have corrected the language (page 23, line 480).

And we modified the description of the association of transcripts with HUA and its nephropathy (page 29, line 510-520).

Table R1: Summary of the expression of genes validated by qRT-PCR in RNA-seq.

AccID Type of gene Log2FC P-Value FDR Style

Adam19 protein-coding 2.156692 2.17E-18 2.96E-16 up

Nlrp12 protein-coding 3.548939 0.000644 0.003986 up

Ptafr protein-coding 1.167864 1.41E-09 4.04E-08 up

A2m protein-coding 6.493011 3.17E-07 5.36E-06 up

LOC102555374 ncRNA 1.64346 0.001121 0.00636 up

LOC102547703 ncRNA 2.199588 4.87E-06 6.02E-05 up

MiR-351-5p ncRNA 1.276317 1.07E-09 2.72E-08 up

Mir-760-5p ncRNA -2.07967 0.002441 0.014917 down

Additional minor criticisms:

Comment 5. For the pathway analysis, authors claim 90 KEGG pathways were enriched with differentially expressed transcripts. Please provide the genes within each of the pathways to determine whether these data represent 90 individual pathways, or whether a small subset of genes in represented in multiple pathways (for example, there could be a small subset of transcripts that could be enriched in several infection related pathways per Figure 4D.)

Response: We have provided genes of each pathway in S4 Table.

Comment 6. Regarding circRNAs, additional explanation is required for how these RNAs were detected. Were these RNAs detected in the RNA-Seq data based on known sequences or were these circRNAs predicted based on the sequencing results. If these transcripts were predicted, pleased provide additional detail as to how these predictions were made, particularly in line 397. Additionally, please provide further explanations of Table 2 and how the ceRNA networks are built. Finally, when introducing circRNAs in lines 93-97, the two sentences are redundant, and one should be eliminated.

The circRNAs were predicted from the reads obtained by sequencing.

Response: We used the special splicing form of circRNA in the expression process to predict the reads obtained from sequencing, and found such a class of reads: covering two exons at the same time and in the opposite direction of the linear RNA, that is, to obtain the possible existence of circRNA in the sequenced sample. More details of the prediction have been added to the manuscript (page 19–20, line 401–408). Table 2 is a summary of circRNA prediction. This table revealed the count, max length, min length and average length of each sample. For the establishment of ceRNA network, firstly, the targeting relationships of significantly DE-miRNAs, DE-circRNA, DE-lncRNA, and DE-mRNA were predicted by miRanda and RNAhybrid (Score < -25), respectively, and the results of the concatenation of the prediction software were taken as the final target gene prediction results. Negative correlation association analysis was performed for miRNA-mRNA, miRNA-circRNA, miRNA-lncRAN according to differential expression type. Finally, miRNA was used as the fit point for the positive correlation joint analysis of circRNA-mRNA and lncRNA-mRNA (page 9–10, line 195–201). Furthermore, excess content about circRNA has been deleted. 

Comment 7. In lines 317-319, authors list the top 5 up- and down-regulated genes. These genes appear to be listed in alphabetical order rather than those that have the greatest fold change or strongest statistical significance. Please update this data based on the volcano plots in Figure 3 – the most extreme fold changes with the lowest FDRs.

Response: Thanks for pointing this out. We are sorry for bothering you with this kind of mistake that we should have avoided and we have updated this date in the Result of revised manuscript (page 15–16, line 321–323).

Comment 8. Please specify which reverse transcriptases were used for qRT-PCR, line 226.

Response: Thank you for your suggestions. And we have added the information of reverse transcriptases in in the Materials and methods of revised manuscript (page 11, line 220–222).

Comment 9. In line 278, authors claim reads mapped to the human genome, but were sequencing rat kidneys. Please correct this typo or provide additional explanation as to why the rat samples map 84.8% to the human genome.

Response: We feel sorry for our carelessness. In our resubmitted manuscript, we have corrected the “mapped to the human genome” into “mapped to the rat genome” in the Result of revised manuscript (page 13, line 281). Thanks for your reminder.

Comment 10. In lines 297-298, authors mention both sets of non-coding RNAs were upregulated. Please specify which set is downregulated.

Response: We apologize for the confusion caused by the incorrect typo. Actually, we intended to clarify the upregulated and downregulated sets of non-coding RNA separately. And we have corrected the mistake that we should avoided in the Result of revised manuscript (page 15, line 301–302). Thanks for your reminder.

Comment 11. Please double check the units for the serum urate measurements, as they are approximately an order of magnitude below reported values (lines 248-249).

Response: We apologize for carelessness, and we have double checked the units then corrected it in the Result of revised manuscript (page 12, line 250–251). Thanks for your suggestion.

Comment 12. Authors mention previous studies in passing but fail to cite references of these studies. Please add citations for lines 102, 132, and 407. Additional citations would be helpful in validating some of the authors’ claims, including lines 63, 71-74, 95, 181, 213, 512-518.

Response: Thanks for the suggestion. We have added the references that the reviewer mentioned and checked the text to assure the claims are verifiable.

Reference

1. Chi K, Geng X, Liu C, Zhang Y, Cui J, Cai G, et al. LncRNA-HOTAIR promotes endothelial cell pyroptosis by regulating the miR-22/NLRP3 axis in hyperuricaemia. J Cell Mol Med. 2021;25(17):8504-21. doi: 10.1111/jcmm.16812. PubMed PMID: 34296520; PubMed Central PMCID: PMC8419175.

2. Xu YT, Leng YR, Liu MM, Dong RF, Bian J, Yuan LL, et al. MicroRNA and long noncoding RNA involvement in gout and prospects for treatment. Int Immunopharmacol. 2020;87:106842. doi: 10.1016/j.intimp.2020.106842. PubMed PMID: 32738598.

3. Dalbeth N, Pool B, Shaw OM, Harper JL, Tan P, Franklin C, et al. Role of miR-146a in regulation of the acute inflammatory response to monosodium urate crystals. Ann Rheum Dis. 2015;74(4):786-90. doi: 10.1136/annrheumdis-2014-205409. PubMed PMID: 25646371.

4. Papanagnou P, Stivarou T, Tsironi M. The Role of miRNAs in Common Inflammatory Arthropathies: Osteoarthritis and Gouty Arthritis. Biomolecules. 2016;6(4). doi: 10.3390/biom6040044. PubMed PMID: 27845712; PubMed Central PMCID: PMC5197954.

5. Li X, Pan Y, Li W, Guan P, You C. The Role of Noncoding RNAs in Gout. Endocrinology. 2020;161(11). doi: 10.1210/endocr/bqaa165. PubMed PMID: 32941616.

6. Ouyang XL, Chen BY, Xie YF, Wu YD, Guo SJ, Dong XY, et al. Whole transcriptome analysis on blue light-induced eye damage. Int J Ophthalmol. 2020;13(8):1210-22. doi: 10.18240/ijo.2020.08.06. PubMed PMID: 32821674; PubMed Central PMCID: PMCPMC7387897.

7. Tang L, Li P, Li L. Whole transcriptome expression profiles in placenta samples from women with gestational diabetes mellitus. J Diabetes Investig. 2020;11(5):1307-17. doi: 10.1111/jdi.13250. PubMed PMID: 32174045; PubMed Central PMCID: PMCPMC7477506.

8. Deng F, Cai L, Zhou B, Zhou Z, Xu G. Whole transcriptome sequencing reveals dexmedetomidine-improves postoperative cognitive dysfunction in rats via modulating lncRNA. 3 Biotech. 2020;10(5):202. doi: 10.1007/s13205-020-02190-9. PubMed PMID: 32309111; PubMed Central PMCID: PMCPMC7154046.

9. He C, Zhao X, Lei Y, Nie J, Lu X, Song J, et al. Whole-transcriptome analysis of aluminum-exposed rat hippocampus and identification of ceRNA networks to investigate neurotoxicity of Al. Mol Ther Nucleic Acids. 2021;26:1401-17. doi: 10.1016/j.omtn.2021.11.010. PubMed PMID: 34900398; PubMed Central PMCID: PMCPMC8636738.

10. Huang J, Ma J, Wang J, Ma K, Zhou K, Huang W, et al. Whole-transcriptome analysis of rat cavernosum and identification of circRNA-miRNA-mRNA networks to investigate nerve injury erectile dysfunction pathogenesis. Bioengineered. 2021;12(1):6516-28. doi: 10.1080/21655979.2021.1973863. PubMed PMID: 34461805; PubMed Central PMCID: PMCPMC8806524.

11. Liu Z, Tan RJ, Liu Y. The Many Faces of Matrix Metalloproteinase-7 in Kidney Diseases. Biomolecules. 2020;10(6). doi: 10.3390/biom10060960. PubMed PMID: 32630493; PubMed Central PMCID: PMCPMC7356035.

12. Zhou D, Tian Y, Sun L, Zhou L, Xiao L, Tan RJ, et al. Matrix Metalloproteinase-7 Is a Urinary Biomarker and Pathogenic Mediator of Kidney Fibrosis. J Am Soc Nephrol. 2017;28(2):598-611. doi: 10.1681/ASN.2016030354. PubMed PMID: 27624489; PubMed Central PMCID: PMCPMC5280025.

13. Hu Q, Lan J, Liang W, Chen Y, Chen B, Liu Z, et al. MMP7 damages the integrity of the renal tubule epithelium by activating MMP2/9 during ischemia-reperfusion injury. J Mol Histol. 2020;51(6):685-700. doi: 10.1007/s10735-020-09914-4. PubMed PMID: 33070277.

14. Yang X, Ou J, Zhang H, Xu X, Zhu L, Li Q, et al. Urinary Matrix Metalloproteinase 7 and Prediction of IgA Nephropathy Progression. Am J Kidney Dis. 2020;75(3):384-93. doi: 10.1053/j.ajkd.2019.07.018. PubMed PMID: 31606236.

15. Higashino T, Morimoto K, Nakaoka H, Toyoda Y, Kawamura Y, Shimizu S, et al. Dysfunctional missense variant of OAT10/SLC22A13 decreases gout risk and serum uric acid levels. Ann Rheum Dis. 2020;79(1):164-6. doi: 10.1136/annrheumdis-2019-216044. PubMed PMID: 31780526; PubMed Central PMCID: PMCPMC6937405.

16. Pham QT, Taniyama D, Sekino Y, Akabane S, Babasaki T, Kobayashi G, et al. Clinicopathologic features of TDO2 overexpression in renal cell carcinoma. BMC Cancer. 2021;21(1):737. doi: 10.1186/s12885-021-08477-1. PubMed PMID: 34174844; PubMed Central PMCID: PMCPMC8236178.

17. Melenhorst WB, van den Heuvel MC, Timmer A, Huitema S, Bulthuis M, Timens W, et al. ADAM19 expression in human nephrogenesis and renal disease: associations with clinical and structural deterioration. Kidney Int. 2006;70(7):1269-78. doi: 10.1038/sj.ki.5001753. PubMed PMID: 16900093.

18. Tattoli F, Falconi D, De Prisco O, Maurizio G, Marazzi F, Marengo M, et al. [Hyperuricemia and gene mutations: a case report]. G Ital Nefrol. 2017;34(3):38-43. PubMed PMID: 28700181.

19. Latchoumycandane C, Nagy LE, McIntyre TM. Myeloperoxidase formation of PAF receptor ligands induces PAF receptor-dependent kidney injury during ethanol consumption. Free Radic Biol Med. 2015;86:179-90. doi: 10.1016/j.freeradbiomed.2015.05.020. PubMed PMID: 26003521; PubMed Central PMCID: PMCPMC4554800.

20. Vlasakova K, Bourque J, Bailey WJ, Patel S, Besteman EG, Gonzalez RJ, et al. Universal Accessible Biomarkers of Drug-Induced Tissue Injury and Systemic Inflammation in Rat: Performance Assessment of TIMP-1, A2M, AGP, NGAL, and Albumin. Toxicol Sci. 2022;187(2):219-33. doi: 10.1093/toxsci/kfac030. PubMed PMID: 35285504.

21. Hansen TB, Jensen TI, Clausen BH, Bramsen JB, Finsen B, Damgaard CK, et al. Natural RNA circles function as efficient microRNA sponges. Nature. 2013;495(7441):384-8. doi: 10.1038/nature11993. PubMed PMID: 23446346.

22. Muhammad, II, Kong SL, Akmar Abdullah SN, Munusamy U. RNA-seq and ChIP-seq as Complementary Approaches for Comprehension of Plant Transcriptional Regulatory Mechanism. Int J Mol Sci. 2019;21(1). doi: 10.3390/ijms21010167. PubMed PMID: 31881735; PubMed Central PMCID: PMCPMC6981605.

23. So A, Thorens B. Uric acid transport and disease. J Clin Invest. 2010;120(6):1791-9. doi: 10.1172/JCI42344. PubMed PMID: 20516647; PubMed Central PMCID: PMCPMC2877959.

---

## [Decision Letter · Decision Letter 1]

9 Sep 2022

PONE-D-21-20162R1Whole transcriptome expression profiles in kidney samples from rats with hyperuricaemic nephropathyPLOS ONE

Dear Dr. Shao,

Your manuscript has now been seen by 4 of the original referees. You will see from their comments below that reviewer #2 continues to raise minor concerns which would need to be addressed. We are quite interested in the possibility of publishing your study in PLOS ONE, but we would like to consider your response to these suggestions in the form of a revised manuscript before we make a final decision on publication.

In addition, we would ask you to address some concerns about figures of the manuscript, as also listed below. Therefore, we invite you to submit a revised version of the manuscript taking into account all reviewer and editor comments. Please highlight all changes in the manuscript text file.

ACADEMIC EDITOR:We would need high quality main figures to represent the data for acceptance of the manuscript. Figure 1A-C can be considered as the labels are bigger. However, currently, it is difficult to read the text of the Figure 2 as well as the labels/legends clearly. This is true for Figure 3A-B and Figure 4 all panels. Figure 5 is hairball but it would be nice if we can get a better resolution image. Figure 6 needs clear labels of the legend and also titles on x and y-axis. Figure 7 legends are totally not readable as well as the text inside the nodes/ circles of the figure. Similar to Figure 4, Figure 8 axis-labels, title and legend need to be made clear. They are difficult to understand even after zooming in. Figure 9 can be accepted as is, but if possible, please make the text labels clear.==============================

We look forward to receiving your revised manuscript.

Kind regards,

Priyadarshini Kachroo

Academic Editor

PLOS ONE

Journal Requirements:

Additional Editor Comments:

Please see the comment above for the figures

Reviewers' comments:

Reviewer's Responses to Questions

**Comments to the Author**

1. If the authors have adequately addressed your comments raised in a previous round of review and you feel that this manuscript is now acceptable for publication, you may indicate that here to bypass the “Comments to the Author” section, enter your conflict of interest statement in the “Confidential to Editor” section, and submit your "Accept" recommendation.

Reviewer #2: All comments have been addressed

2. Is the manuscript technically sound, and do the data support the conclusions?

Reviewer #2: Yes

3. Has the statistical analysis been performed appropriately and rigorously? 

Reviewer #2: N/A

4. Have the authors made all data underlying the findings in their manuscript fully available?

Reviewer #2: Yes

5. Is the manuscript presented in an intelligible fashion and written in standard English?

Reviewer #2: Yes

6. Review Comments to the Author

Reviewer #2: I appreciate authors' effort for updating the manuscript but have a few remaining suggestions.

Regarding comment 1, how about displaying the data as a bar plot with dots? And it would be better to adjust the range of y axis for highlighting the differences.

For example, y axis could be narrowed down from 0.8 to 1.5 for Nlrp12.

Regarding comment 3, the authors modified the explanation in page 27, line 563-565 as follows.

"PAF is a 562 phospholipid that plays an important role in tumour transformation, tumour growth, angiogenesis, 563 metastasis and pro-inflammatory processes[63]. Haemorrhagic fever[64] and bacterial sepsis[65]"

As far as I understand, "Haemorrhagic fever[64] and bacterial sepsis[65]" is not a sentence, so it is hard to get what it means.

7. PLOS authors have the option to publish the peer review history of their article (what does this mean?). If published, this will include your full peer review and any attached files.

Reviewer #2: No

---

## [Author Response · Author response to Decision Letter 1]

30 Sep 2022

Dear Editor and Reviewers,

Thank you for taking time out of your busy schedule to review the manuscript again. These comments are all valuable and helpful for improving our manuscript, as well as the important guiding significance to our research. We have studied comments carefully and have made correction which we hope meet with approval. And we use the “Track Changes” option in Microsoft Word to show the revised portions, deleted content is shown with a red strikeout and added content is shown with a red underscore. The revision instructions are as follows:

Summary of the revision：

Discussion: We removed the unreasonable parts to make the sentence more coherent, and also deleted their references accordingly.

Figure: We have made changes to all figures to make text labels and legends clearer. The range of the Y axis has also been adjusted to highlight the differences.

Responses to the reviewers’ and editorial comments 

(Response: answers from the authors are in blue color, the figures that are just used in the response are named Figure R+number or Table R+number)

Priyadarshini Kachroo

Academic Editor

Comment 1. We would need high quality main figures to represent the data for acceptance of the manuscript. Figure 1A-C can be considered as the labels are bigger. 

Response: Thanks for your comments. We have carefully revised the figure.

Comment 2. However, currently, it is difficult to read the text of the Figure 2 as well as the labels/legends clearly. 

Response: Thank you for your significant reminder. The pictures have been corrected.

Comment 3. This is true for Figure 3A-B and Figure 4 all panels. 

Response: Thank you for pointing this out. We checked the figures and modified the label/legend of the figures.

Comment 4. Figure 5 is hairball but it would be nice if we can get a better resolution image. 

Response: We have uploaded the figures and adjusted the resolution of figure to 600 dpi according to the requirement of the Plos one.

Comment 5. Figure 6 needs clear labels of the legend and also titles on x and y-axis.

Response: Thanks for the reminder. We modified the legend labels in Figure 6 and the titles on the x- and y-axis to make them clearer.

Comment 6. Figure 7 legends are totally not readable as well as the text inside the nodes/ circles of the figure. 

Response: We apologize that our figures are unreadable. And we have modified Figure 7 based on the editor's suggestion.

Comment 7. Similar to Figure 4, Figure 8 axis-labels, title and legend need to be made clear. They are difficult to understand even after zooming in.

Response: Thank you for your valuable comments. We modified the axis labels, title and legend in Figure 8.

Comment 8. Figure 9 can be accepted as is, but if possible, please make the text labels clear.

Response: Thank you for your advice. We have modified the text labels in Figure 9.

Reviewer #2: 

Comment 1. Regarding comment 1, how about displaying the data as a bar plot with dots? And it would be better to adjust the range of y-axis for highlighting the differences.

For example, y axis could be narrowed down from 0.8 to 1.5 for Nlrp12.

Response: Thank you for pointing this out. According to your requirements, we have reduced the y-axis, shrunk the y-axis of Figures 9B, C, and H from 1.5 to 1.2, and the y-axis of Figure 9G from 2 to 1.8, making the difference look more obvious in the figure.

Comment 2. Regarding comment 3, the authors modified the explanation in page 27, line 563-565 as follows.

"PAF is a 562 phospholipid that plays an important role in tumour transformation, tumour growth, angiogenesis, 563 metastasis and pro-inflammatory processes[63]. Haemorrhagic fever[64] and bacterial sepsis[65]"

As far as I understand, "Haemorrhagic fever[64] and bacterial sepsis[65]" is not a sentence, so it is hard to get what it means.

Response: We are sorry for our carelessness and we have corrected the mistakes in the revised manuscript (page 27, line 563) and adjusted the order of the references. Thanks for your reminder.

---

## [Editor Report · Decision Letter 2]

11 Oct 2022

Whole transcriptome expression profiles in kidney samples from rats with hyperuricaemic nephropathy

PONE-D-21-20162R2

Dear Dr. Shao,

We’re pleased to inform you that your manuscript has been judged scientifically suitable for publication and will be formally accepted for publication once it meets all outstanding technical requirements.

Kind regards,

Priyadarshini Kachroo

Academic Editor

PLOS ONE
---

## [Editor Report · Acceptance letter]

6 Dec 2022

PONE-D-21-20162R2 

Whole transcriptome expression profiles in kidney samples from rats with hyperuricaemic nephropathy 

Dear Dr. Shao:

I'm pleased to inform you that your manuscript has been deemed suitable for publication in PLOS ONE. Congratulations! Your manuscript is now with our production department. 

Kind regards, 

on behalf of

Dr. Priyadarshini Kachroo 

Academic Editor

PLOS ONE